# Linear mapping approximation of gene regulatory networks with stochastic dynamics

Zhixing Cao [1] & Ramon Grima[1]

The presence of protein–DNA binding reactions often leads to analytically intractable models of stochastic gene expression. Here we present the linear-mapping approximation that maps systems with protein–promoter interactions onto approximately equivalent systems with no binding reactions. This is achieved by the marriage of conditional mean-field approximation and the Magnus expansion, leading to analytic or semi-analytic expressions for the approximate time-dependent and steady-state protein number distributions. Stochastic simulations verify the method's accuracy in capturing the changes in the protein number distributions with time for a wide variety of networks displaying auto- and mutual-regulation of gene expression and independently of the ratios of the timescales governing the dynamics. The method is also used to study the first-passage time distribution of promoter switching, the sensitivity of the size of protein number fluctuations to parameter perturbation and the stochastic bifurcation diagram characterizing the onset of multimodality in protein number distributions.

[1] School of Biological Sciences, the University of Edinburgh, Mayfield Road, Edinburgh EH9 3JH Scotland, UK. Correspondence and requests for materials should be addressed to R.G. (email: ramon.grima@ed.ac.uk)

Gaining detailed quantitative insight into the dynamics of single living cells is one of the main goals of modern molecular biology. It is well acknowledged that a systems biology approach, whereby alternating cycles of mathematical modeling and experiments lead to refined understanding of the biological system is ideal[1]. In such an approach, the prediction of a mathematical model is contrasted with experimental data: a good match implies that the model offers a potential explanation of the observations (and potentially an estimation of the parameters) while a bad match implies that further refinement of the model (and probably further experiments) is necessary. The output of the experiments is often the number of fluorescently tagged proteins as a function of time from which one can calculate the probability distribution and its associated moments such as the mean and variance in the protein numbers. Clearly then a mathematical model is useful in this systems biology approach to living cell dynamics, if it can accurately predict the distribution of protein numbers and herein lies a problem: exact solutions of the stochastic description of gene regulatory networks (GRNs) have only been reported for a few simple cases. In this article, we describe a novel method which circumvents the aforementioned problem by deriving approximate but accurate solutions to the probability distributions of protein numbers of a wide variety of GRNs.

Before we describe the method, we summarize the state of the art in the mathematical modeling of GRNs. It is well known that such networks suffer from noise principally due to the low copy number of genes, mRNA and of some protein molecules inside single cells[2,3]. Hence, a stochastic mathematical framework is necessary to describe the dynamics of GRNs. The accepted modern-day framework is the Chemical Master Equation (CME), which is a set of differential equations describing the probabilistic evolution of states of the GRN[4]. Exact solutions of the CME have only been reported for a few simple GRNs: (i) the time-dependent solution of the CME of a GRN involving the reversible switching between two promoter states, the production of mRNA by the active state and the degradation of mRNA[5]; (ii) the time-dependent solution of the CME of a GRN involving the transcription of mRNA by an active promoter, the translation of the mRNA into protein and the decay of both protein and mRNA[6]; (iii) the steady-state solution of a GRN of a negative or positive feedback loop, whereby a promoter can produce proteins with a certain rate in the inactive state and with a different rate in the active state and it switches from the inactive to the active state by binding a protein molecule. This model also includes protein degradation[7]; (iv) the same as in (iii) but with the production rate occurring in bursts[8]. In models (i) and (ii), every reaction is either zero or first-order and hence we shall refer to these as linear GRNs since by the law of mass action, the rate of every reaction is linear in the concentrations. We shall refer to models (iii) and (iv) as nonlinear GRNs because there is a second-order reaction involving the binding of protein to the promoter, whose rate is nonlinear in the concentrations. Note that the exact time-dependent solution has only been obtained for linear GRNs; for the nonlinear GRNs only the steady-state solution is known. It is also the case that none of these model an external time-varying stimulus to the GRN, a commonly observed feature, e.g., circadian clocks.

Notwithstanding these difficulties, some have devised methods to obtain expressions for the approximate probability distribution solutions of the CME for nonlinear GRNs under various assumptions: (i) the fluctuations in copy numbers are very small and the distribution is Gaussian[9–11]; (ii) that there exists time-scale separation, e.g. slow promoter switching[12–16]; (iii) the promoter states are uncorrelated[17]; (iv) the volume of the cell is large enough that the CME can be approximated by a few terms in the system-size expansion[18]. All of these methods generate approximate time evolving distributions for GRNs with second-order reactions (for a comprehensive recent review see[19]) and hence circumvent the issues of exact solutions of the CME. The disadvantages of these methods are however considerable because of their limiting assumptions: (i) distributions measured in vivo are often highly skewed and sometimes multimodal, i.e. non-Gaussian; (ii) timescale separation occurs in a few cases but is not generally the case in nature; (iii) promoter states are often correlated due to the presence of feedback loops; (iv) it is impossible to a priori estimate how many terms are needed in the system-size expansion to obtain an accurate result. There are also methods which compute the approximate distribution numerically without explicit analytical expressions (see for e.g.[20–23]); of these the Stochastic Simulation Algorithm (SSA)[20] is of particular importance because the approximation error is equal to the sampling error and hence can be made arbitrarily small.

In this article, we devise a novel type of approximate solution of the CME which provides analytical or semi-analytical expressions for the time-dependent and steady-state solution of common nonlinear GRNs without making a priori assumptions on the form of the distribution or invoking timescale separation and which is even applicable to GRNs with an external time-varying stimulus.

## Results

**Illustrating the linear mapping approximation by an example.** The solution of the CME of linear GRNs is typically easier than the solution of the CME of nonlinear GRNs. This observation leads to the question: is it possible to map, in an approximate way, a nonlinear GRN onto an equivalent linear GRN such that the exact solution of the latter gives an approximate solution of the former?

We shall first develop the method on a simple nonlinear feedback loop which is schematically shown in Fig. 1a (upper). A promoter switches between two states $G$ and $G^*$, and each state is associated with a different rate of protein production. The switch from $G$ to $G^*$ occurs through the binding of a protein molecule to $G$ and the protein can also decay. This is a rudimentary form of a feedback loop: if $\rho_u > \rho_b$, then it functions as a negative-feedback loop (protein represses its own expression) and otherwise it is a positive feedback loop (protein activates its own expression). Here we do not explicitly model the mRNA for simplicity purposes. This nonlinear GRN can be transformed in to a linear GRN (Fig. 1a lower) by removing the second-order reaction between protein and state $G$. Specifically, we replace the reversible reaction $G + P \rightleftharpoons G^*$ by $G \rightleftharpoons G^*$. Note that all parameters between the two models are the same except for $\sigma_b$ and $\bar{\sigma}_b$. The question now is: given the nonlinear GRN with a certain set of parameters, how can we select the free parameter $\bar{\sigma}_b$ in the linear GRN such that the solution of this system well approximates the solution of the original nonlinear GRN?

First we write the exact moment equations for the linear GRN (which can be straightforwardly obtained from the CME—see Methods):

$$\partial_t \langle n_p \rangle = \rho_u \langle n_g \rangle + \rho_b \left( 1 - \langle n_g \rangle \right) - \langle n_p \rangle,$$
$$\partial_t \langle n_g \rangle = -\bar{\sigma}_b \langle n_g \rangle + \sigma_u \left( 1 - \langle n_g \rangle \right), \qquad (1)$$
$$\partial_t \langle n_p n_g \rangle = \rho_u \langle n_g \rangle + \sigma_u \langle n_p \rangle - (1 + \bar{\sigma}_b + \sigma_u) \langle n_p n_g \rangle,$$

where $\partial_t$ denotes the time derivative, $n_p$ is the number of molecules of protein $P$, $n_g$ is a Boolean variable taking the value of 1 if the promoter is in state $G$ and the value 0 if it is in state $G^*$

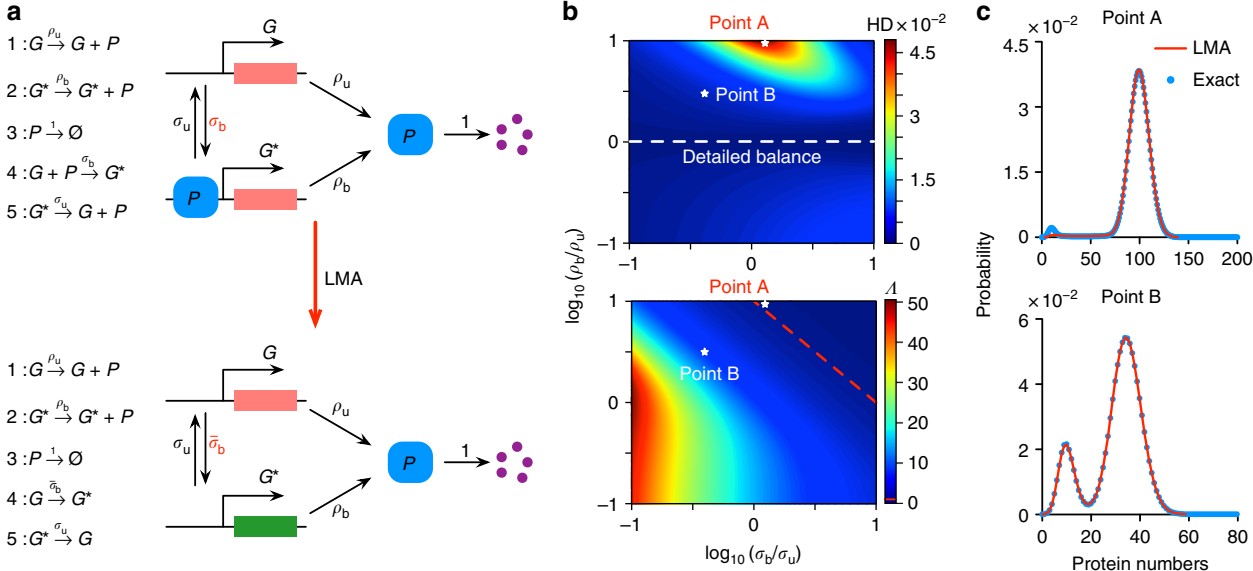

**Fig. 1** Linear mapping approximation (LMA) and its application to steady-state conditions. **a** Illustration of the main idea behind the LMA namely to approximate the reversible (nonlinear) reaction between protein and promoter in the nonlinear GRN by a first-order (linear) reaction with an effective reaction rate in a linear GRN. **b** The upper figure shows a heatmap of Hellinger distance (HD) between the LMA and the exact probability distribution of protein numbers in steady-state conditions with parameters $\rho_u = 10$, $\sigma_u = 0.01$ for the nonlinear GRN shown in **a**. The exact distribution is reported in[7]. The bottom figure shows a heatmap of $\Lambda$, which is the ratio of the values of the two eigenvalues of the Jacobian of the deterministic rate equations of the nonlinear GRN in steady-state conditions. The red broken line denotes the contour line of $\Lambda = 1$. Note that the value of the HD is very small over a wide range of the ratio of time scales $\Lambda$ indicating that the LMA's accuracy is independent of time-scale separation. **c** A comparison of the LMA and exact steady-state distributions for Points A and Point B, marked as white stars, on the heatmap in **b**; note that Point A corresponds to the parameter set with the largest HD ($\rho_b = 100$, $\sigma_b = 0.0126$ with HD of 0.0478). Point B corresponds to $\rho_b = 35$, $\sigma_b = 0.004$ with HD of 0.0032

and bracket $\langle \cdot \rangle$ is the expection operator. Next we note that the first-order reaction $G \rightarrow G^*$ in the linear GRN maps onto the second-order reaction $G + P \rightarrow G^*$ if we select $\bar{\sigma}_b = \sigma_b \left( n_p | n_g = 1 \right)$ where $n_p | n_g = 1$ is the instantaneous number of proteins $n_p$ given the promoter is in state $G$. The simplest approximation is to use the expectation value of this stochastic rate such that we have:

$$\bar{\sigma}_b = \sigma_b \left\langle n_p | n_g = 1 \right\rangle = \sigma_b \frac{\left\langle n_p n_g \right\rangle}{\left\langle n_g \right\rangle}. \quad (2)$$

This is a mean-field assumption and is expected to be accurate when the size of the fluctuations in the number of proteins, given the promoter is in state $G$, are small compared to the mean number of proteins conditional on the same state. Since experiments show that the standard deviation of the fluctuations divided by the mean molecule number roughly scales as the inverse square root of the mean molecule number[24], it follows that the mean-field assumption should be accurate provided the mean protein molecule numbers in state $G$ are not too small.

Substituting Eq. (2) in Eq. (1) and solving the resultant coupled set of differential equations, we obtain a time-dependent solution for the moments. These solutions can then be substituted in Eq. (2) to obtain our estimate for the effective rate parameter in the linear (mapped) GRN:

$$\bar{\sigma}_b = f \left( t, \rho_u, \rho_b, \sigma_u, \sigma_b \right), \quad (3)$$

where $f$ denotes function of. This function can generally be obtained by numerical solution of the aforementioned modified differential equations; in steady-state conditions an explicit

formula can also be obtained:

$$\bar{\sigma}_b = \frac{-1 + \rho_b \sigma_b - \sigma_u + \sqrt{\left( 1 - \rho_b \sigma_b + \sigma_u \right)^2 + 4 \rho_u \sigma_b (1 + \sigma_u)}}{2}. \quad (4)$$

The last and remaining question is how can we use this parameter estimate to build the full time-dependent solution of the nonlinear GRN. We observe that the time-dependent probability distribution solution of the CME of the linear GRN with general time-dependent $\bar{\sigma}_b$ is likely impossible to obtain in closed-form. However it is possible to solve if $\bar{\sigma}_b$ were a constant independent of time (see "Methods" section); let this general probability distribution solution be denoted as $\mathcal{S}_{FLTD}(\bar{\sigma}_b, t)$. We shall then make the assumption that the time-dependent probability distribution solution of the CME of the linear GRN with general time-dependent $\bar{\sigma}_b$ given by Eq. (3) is well approximated by $\mathcal{S}_{FLTD}(\bar{\sigma}_b^*, t)$ where $\bar{\sigma}_b^*$ is the time-average of Eq. (3):

$$\bar{\sigma}_b^* = \frac{\int_0^t f \left( t', \rho_u, \rho_b, \sigma_u, \sigma_b \right) dt'}{t}. \quad (5)$$

A rigorous theoretical justification of this assumption can be found in the Methods. Hence the linear mapping approximation (LMA) of the probability distribution of the nonlinear feedback loop at time $t$ is given by $\mathcal{S}_{FLTD}(\bar{\sigma}_b^*, t)$. Note that the time-averaging assumption is only needed if one wants to calculate the distribution in finite time; in steady-state, there is no need of the assumption since then $\bar{\sigma}_b$ is constant (and equal to Eq. (4)) and the steady-state probability distribution is directly given by $\mathcal{S}_{FLTD}(\bar{\sigma}_b, t \rightarrow \infty)$.

To summarize, the LMA procedure to find the approximate time-dependent probability distribution of protein numbers at time $t$ in a general nonlinear GRN involves the following steps: (i)

find the linear GRN by replacing any reversible promoter-protein reaction in the nonlinear GRN by a reversible pseudo first-order reaction between promoter states with stochastic rates; (ii) write the closed-set of moment equations for the linear GRN with the stochastic rates replaced by their means, solve for the moments at time $t$ and use the latter to obtain the approximate value of the rate parameter/s at time $t$ characterizing the pseudo first-order reaction/s in the linear GRN; (iii) calculate the time-average of these parameters over the time interval $[0, t]$; (iv) obtain the time-dependent probability distribution solution of the CME of the linear GRN assuming the rate parameter/s characterizing the pseudo first-order reactions are time-independent constants; (v) the approximate time-dependent probability distribution of the nonlinear GRN at time $t$ is then given by replacing the "constant" rate parameter/s characterizing the pseudo first-order reactions solution in step (iv) by the time-averaged parameters calculated in step (iii).

Steps (i) to (iii) can always be performed but steps (iv) and (v) require the existence of a closed-form solution for the linear GRN and this is the major limitation of the method. When such a solution exists, then for a nonlinear GRN with $N$ protein–promoter binding reactions, the approximate time-dependent probability distribution given by the LMA is a closed-form distribution with $N$ effective parameters to be determined numerically. In practice, this leads to a considerable computational advantage over purely numerical methods such as the SSA[20] and the Finite State Projection method[21] (see Supplementary Note 3 for details) simply because the closed-form distribution is composed of well-known functions that can be evaluated by standard symbolic packages in fractions of a second.

If one is only interested to find an approximate steady-state probability distribution of protein numbers then the procedure is considerably simpler. Step (i) is as before. Step (ii) is the same but now the moments are found in steady-state. The final approximate solution is then obtained by substituting the effective rate parameters found in Step (ii) in the steady-state probability distribution solution of the CME of the linear GRN. In many cases, these steps can be done analytically and hence the output is an approximate solution in closed-form.

Note that independent of whether we are interested in the time-dependent or steady-state problem, when a closed-form solution for the linear GRN does not exist, the method still gives approximate expressions for all the moments of the nonlinear GRN using steps (i) and (ii); in this case, its output is similar to moment-closure methods (see ref. [19] for a recent review) but with the advantage that we have made no implicit assumption on the form of the probability distribution solution of the chemical master equation.

**The LMA of common nonlinear GRNs.** Next we will test the accuracy of this method for various nonlinear GRNs using both exact results and stochastic simulations. In particular, we want to clearly show that the LMA accurately predicts probability distributions for protein numbers which are unimodal or bimodal, Gaussian or skewed, in steady-state or evolving in time and independent of timescale separation.

In Fig. 1, we show the high accuracy of the LMA in predicting the probability distribution of protein numbers for the feedback loop in steady-state conditions. In particular, Fig. 1b (upper) shows a heat map of the Hellinger distance (HD) between the exact steady-state probability distribution of the nonlinear GRN (reported in[7]) and the approximate probability distribution given by the LMA (as described earlier). Note that the HD has the properties of being symmetric and satisfies the triangle inequality,

thus implying that it is a distance metric on the space of probability distributions (unlike for example the commonly used Kullback–Leibler divergence). Since it returns a number between 0 and 1, it is clear that the distance between the exact and approximated distributions is very small for both negative ($\rho_b < \rho_u$) and positive feedback ($\rho_b > \rho_u$). This is further confirmed by explicitly showing in Fig. 1c, the distribution for two points in the heat map in Fig. 1b (upper): the LMA distribution with the largest Hellinger distance (Point A) is barely noticeably different from the exact distribution and the LMA does extremely well even when the distribution is bimodal (Point B). It can also be easily proved that the LMA distribution is exact when the system is in detailed balance conditions. This is since in such conditions, $\rho_u = \rho_b$[7], which implies that the protein distribution is unaffected by the bimolecular reaction at the heart of promoter switching and hence the system acts as a linear GRN in this special case. In Fig. 1b (lower), we further confirm the hypothesis that the LMA does well in steady-state conditions independent of the existence of timescale separation conditions: a comparison of the heat plots in Fig. 1b upper and lower shows that while the ratio of the gene and protein timescales ($\Lambda$) varies considerably (0.1 to 50) over the region of parameter space considered, there is very little corresponding change in the HD (0 to $4.5 \times 10^{-2}$). There is also no correlation between the two heat plots. $\Lambda$ is the ratio of the two eigenvalues obtained from the Jacobian of the deterministic rate equations. Hence to sum up, in steady-state the LMA predictions for the feedback loop are accurate independent of the type of feedback (positive or negative), modality of the distribution and of timescale separation conditions.

Next we test the accuracy of the LMA for predicting the time-evolution of the probability distribution of proteins in four common types of nonlinear GRNs (or motifs): the feedback loop (Fig. 1a upper), the feedback loop with protein bursting (Fig. 2a), the feedback loop with cooperative protein binding (Fig. 2b) and the feedback loop with oscillatory transcription rates (Fig. 2c). Details of these loops, their master equation formulation and corresponding LMA can be found in Methods and the Supplementary Information. Note that the decay rate of proteins in all cases is set to unity; this is not an arbitrary choice but rather stems from the fact that the time in the master equation can always be non-dimensionalised using the actual value of the protein decay rate $k_d$. Hence all times shown in the graphs should be understood to be non-dimensional and equal to the real time multiplied by $k_d$ while all other parameters ($\rho_u$, $\rho_b$, $\sigma_u$, $\sigma_b$) should be understood to also be non-dimensional and equal to the real value of the parameter divided by $k_d$. Bursting (production of proteins in bursts), cooperativity (multiple proteins binding the promoter) and time-varying transcription rates are common features observed in many GRNs in both eukaryotic and prokaryotic cells (see for example ref. [25–27]). Note that an implicit description of mRNA exists in the model with protein bursting because protein burst sizes distributed according to a geometric distribution are obtained when the protein is produced by a fast intermediate mRNA, a common scenario in bacteria and yeast[15]. Note also that while all the three systems are composed of reactions with mass-action propensities, in certain limits they reduce to systems composed of effective reactions with non-mass action propensities e.g. under quasi-equilibrium conditions between promoter and protein, the model of a feedback loop with cooperative binding reduces to an effective model describing protein production with a Hill-type propensity[28,29].

Figure 2d shows that in all cases the LMA distribution agrees very well with that obtained from stochastic simulations using the SSA[20]. In particular the LMA precisely captures the change in shape of the distribution with time from Gaussian at short times to a skewed unimodal distribution at intermediate times to

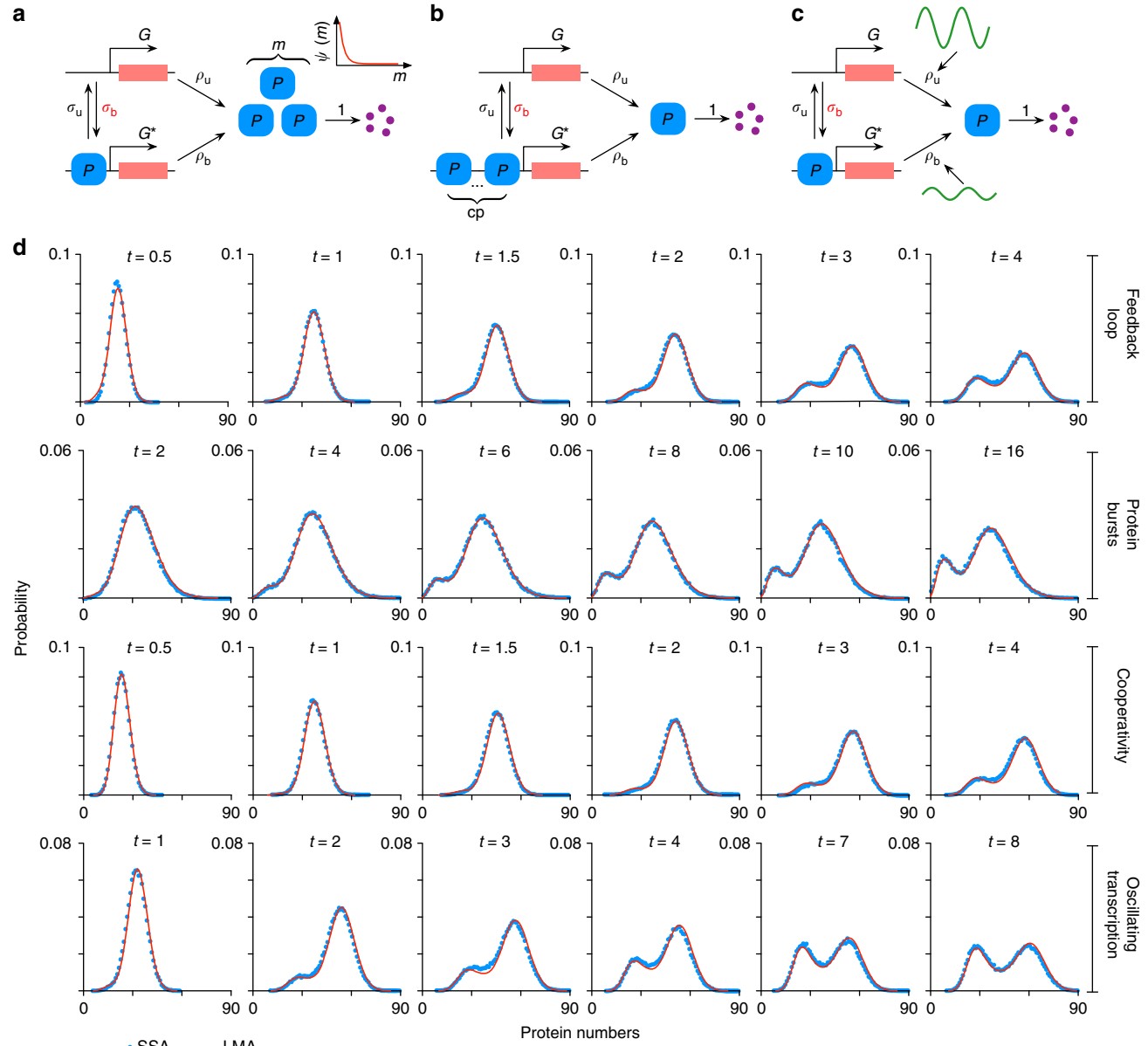

**Fig. 2** LMA approximation for the time-dependent probability distribution of protein numbers for various nonlinear GRN. The feedback loop shown in Fig. 1a upper, a feedback loop with protein bursting (**a**), a feedback loop with cooperativity (**b**) and a feedback loop with oscillatory transcription (**c**). The inset of **a** shows the probability distribution $\psi(m)$ of the protein burst size $m$: we consider a geometric distribution, the discrete analog of the exponential distribution which has been measured in experiments[15, 25]. The number of proteins binding to the promoter for the cooperative loop in **b** is given by cp. In **d**, we show snapshots of the protein number distribution at various times. The LMA approximation (red line) agrees with the results of stochastic simulations using the SSA (blue dots), and is able to capture the transition from unimodality at short times to bimodality at long times. The parameters are: for feedback loop $\rho_u$ = 60, $\rho_b$ = 25, $\sigma_b$ = 0.004, $\sigma_u$ = 0.25; for protein bursts $\rho_u$ = 20, $\rho_b$ = 5, $\sigma_b$ = $10^{-3}$, $\sigma_u$ = 0.1 and the mean burst size $b$ = 2; for cooperativity $\rho_u$ = 60, $\rho_b$ = 25, $\sigma_b$ = 5 × $10^{-5}$, $\sigma_u$ = 0.25 and cooperativity order cp = 2; for oscillating transcription, the parameters are $\rho_u$ = 60, $\rho_b$ = 25, Am = 0.3, $k$ = 1.33, $\sigma_b$ = 0.004, $\sigma_u$ = 0.25. The SSA result in each snapshot is obtained by averaging over 80,000 realizations and the sampling error is <2% for each point in the SSA probability distribution. In all cases, the initial conditions are zero protein in promoter state $G$

bimodal at long times. Furthermore for the oscillatory transcription feedback loop, it can be shown that the LMA correctly captures the oscillatory nature of the mean and variance in protein numbers and accurately predicts the phase difference between the oscillations in the mean protein numbers and in the transcription rate (see Supplementary Fig. 1).

Next, we seek to understand the dependence of the error in the LMA predictions with parameter values and the intuitive reasons underlying such relationships. In Fig. 3a, we show the HD (between the SSA calculated distribution and the LMA distribution) as a function of time for the feedback loop with cooperative protein binding with parameters $\rho_u$ = 60, $\rho_b$ = 25 and for various values of $\sigma_b$. While there is no apparent relationship between HD and $\sigma_b$ in steady-state, it is clear that the maximum of the HD (over time) increases with $\sigma_b$. The corresponding probability distributions for these three maxima (A, B and C) are shown in Fig. 3c upper. The degree of nonlinearity in the GRN is controlled by the rate $\sigma_b$ of the only nonlinear (second-order) reaction in the nonlinear GRN and hence one would expect our approximate linear mapping to be less accurate as $\sigma_b$ increases

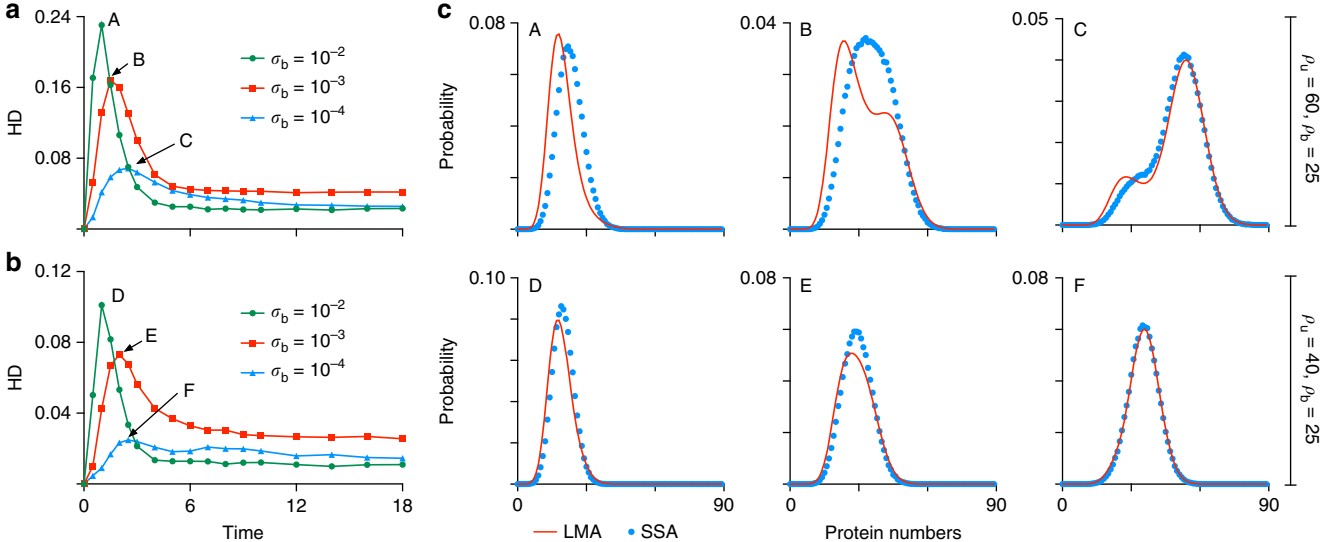

**Fig. 3** Dependence of the LMA approximation error on parameters in non-oscillatory feedback loops. **a**, **b** Show the variation of the Hellinger distance (HD) between LMA and SSA distributions for protein numbers in the cooperative feedback loop as a function of time, $\sigma_b$ and $\rho_u - \rho_b$. The results show that the HD reaches its maximum at intermediate times and that this value is an increasing function of $\sigma_b$ (the rate parameter controlling the degree of nonlinearity in the nonlinear GRN) and of the difference in gene expression between the two promoter states ($\rho_u - \rho_b$). The other parameters are: cp = 2, $\sigma_u = 0.25$. **c** Compares the LMA predictions and SSA distributions for six time points (at which the HD maximizes) as indicated in **a**, **b**

(for a more precise explanation see the subsection on the justification of the time-averaging assumption in the "Methods" section). In Fig. 3b, we repeat the same analysis but now using parameters $\rho_u = 40$, $\rho_b = 25$. The same relationship is seen between the maximum of the HD (over time) and $\sigma_b$ however the absolute values of the error are now reduced by about half. This can be explained due to the fact that the difference between $\rho_u$ and $\rho_b$ is smaller in this case than for the one shown in Fig. 3a and we already know that as $\rho_u$ approaches $\rho_b$, the bimolecular reaction behind promoter switching becomes irrelevant and the LMA becomes exact. We also note that in all cases, the maximum HD was obtained at intermediate times (rather than in steady-state). This is since the HD must be zero initially since we start with the same initial conditions in the LMA and SSA, it must be significant in finite time because both assumptions (mean-field and time averaging) are being used while it must be small in steady-state because only the mean-field assumption is being used. These results are typical of the three nonlinear GRN studied (feedback loop, feedback loop with bursting and feedback loop with cooperativity) which all possess non-zero, non-oscillating moments of protein molecule numbers at steady-state. In summary, the error in the LMA prediction is typically smaller in steady-state compared to time-evolution and it achieves a maximum whose value increases with the rate parameter controlling the nonlinear protein–promoter binding reaction and with the difference between the protein production rates of the two promoter states.

We studied also the error in the LMA predictions for the feedback loop with oscillatory transcription which leads to oscillating moments in the protein numbers in steady-state and hence is in a different class than the previous three nonlinear GRNs. One can think of this nonlinear GRN as an oscillating input signal passing through a filter (composed by the interacting molecular components) with output given by the mean protein numbers. Depending on the type of filter, one would expect certain frequencies will be more attenuated than others. Indeed this is what is observed in a plot (Fig. 4) of the amplitude in the oscillations in the mean protein numbers as a function of the frequency of the oscillating transcription (the input frequency).

The SSA predicts the amplitude to gently peak at a frequency of about 0.2; the rate equations predict the same albeit with a different protein amplitude. The LMA however does not capture the peak and simply predicts a decreasing amplitude with increasing frequency which agrees very well with the SSA for frequencies larger than that of the peak. In fact it can be proved (see Supplementary Note 2 Eq. (12)) that independent of parameter values and of the amplitude and frequency of the oscillatory transcription, the LMA predicts the amplitude of the mean protein oscillations to decrease monotonically with increasing frequency. This behavior of the LMA is due to its time-averaging assumption: the amplitude of the time-average of a sinusoidal function is inversely proportional to the frequency. In summary the accuracy of the LMA's predictions is likely low for input frequencies close to the intrinsic resonant frequency of a general nonlinear GRN and high otherwise.

The master equations studied thus far have been limited to GRNs with two promoter states, reactions with mass-action propensities and no explicit description of mRNA. While an implicit description of mRNA exists in the feedback loop with protein bursting model, an explicit description has the advantage that it gives information about both mRNA and protein and can hence can be useful to interpret experiments producing such type of data (see for example ref. [30]). Also we earlier mentioned that an implicit description of effective non-mass action propensities of the Hill-type exists in the model of a feedback loop with cooperativity; an explicit description in the sense of using directly Hill-type propensities in the master equation can sometimes be helpful when we want to work with a reduced model in terms of few parameters. In Figs. 5 and 6, we show the application of the LMA to master equations describing systems with more than two promoter states, effective reactions with non mass-action propensities and including mRNA dynamics. Specifically we find that the LMA accurately captures: (1) the time-evolution of the protein distributions for the 4 promoter state toggle switch (Fig. 5a, b), which involves the expression and mutual repression of two different proteins $P$ and $M$; (2) the steady-state protein distribution in a two promoter feedback loop where the protein decays via the (non-mass action) Michaelis–Menten like

propensity function (Fig. 5c, d); (3) the mRNA steady-state distribution in a two promoter feedback loop which models both mRNA transcription and protein translation (Fig. 6a, b). Details of the LMA for these three systems can be found in the Supplementary Notes 5–7.

Note that for the two promoter feedback loop modeling transcription and translation, the mRNA distribution can also be computed in time; however, the protein distribution cannot currently be obtained from the LMA in time or steady-state. The reason is that the LMA maps this GRN on to a linear network (also called three-stage gene expression in ref. [15]), for which there is an exact analytical solution for the marginal distribution of mRNA numbers but no solution is currently known for the marginal distribution of protein numbers. However, note that nevertheless the LMA does give all the moments of the protein distribution and these are shown in Fig. 6c to be very accurate compared to those obtained from the SSA, independent of the ratio of the timescales of protein and mRNA—this is particularly relevant to the description of mammalian gene expression[31] where the ratio of timescales varies widely. Analytical solutions for the linear network are known for the case of timescale separation of protein and mRNA lifetimes[15] (conditions compatible with gene expression in bacteria and yeast) and thus in this case by use of the LMA, one can obtain the corresponding analytical solutions for both the mRNA and protein marginal distributions for the feedback loop shown in Fig. 6a.

**Further applications of the LMA.** Having verified the high accuracy of the LMA, we shall next use it to shed light on how the stochastic properties of a feedback loop are affected by cooperativity and protein bursting. In particular we are interested in how these two features affect: the first-passage time distribution of switching from one promoter state to the other, the sensitivity of the coefficient of variation squared to a change in the parameter values and the stochastic bifurcation diagram.

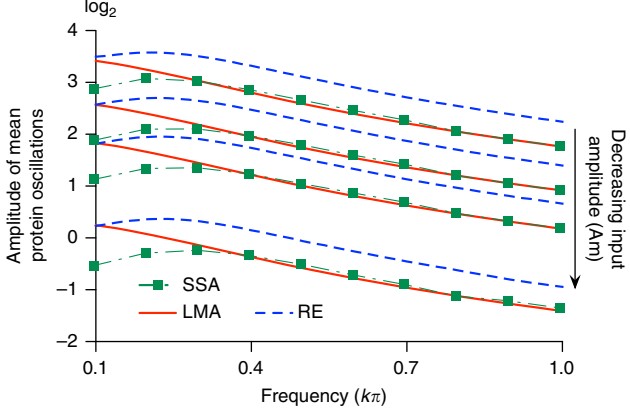

**Fig. 4** Dependence of the LMA approximation error on parameters in the oscillatory feedback loop. The figure shows a plot of the amplitude of the oscillations in mean protein number as a function of the frequency of the oscillatory transcription ($k\pi$) and of its amplitude (Am). This is obtained using the LMA (red solid line), the SSA (green squares) and the deterministic rate equations (RE, dashed blue line). The results shows that the rate equations cannot predict the amplitude precisely but capture the weak resonance phenomenon while the LMA gives precise predictions at high frequencies, but cannot capture the resonance. The frequency selectivity of the LMA is due to its time-averaging assumption. The oscillatory feedback loop is the one shown in Fig. 2c with $\rho_u = 20$, $\rho_b = 0$, $\sigma_b = 0.04$, $\sigma_u = 0.25$, and Am is selected to be 0.9, 0.5, 0.3, 0.1 (from top to bottom). Note that the transcription rate in the active promoter state is $\rho_u[1 + Am\cos(k\pi t)]$

Practically, all GRNs involve multiple promoter states with different post-translational pathways enabled by each state. Hence the switching from one state to another is important to understand from the perspective of cellular decision-making, e.g., a cell's response to a stimulus may require the quick switching on of certain biochemical machinery. This can be mathematically characterized using the first-passage time (FPT) distribution which is the probability distribution of the time it takes to switch between two promoter states given initially one of the states. The switch from $G^*$ to $G$ occurs via $G^* \rightarrow G + P$, which is a linear reaction with rate $\sigma_u$ and hence it can be easily shown that the FPT for the promoter switching from state $G^*$ to $G$ is simply exponential distribution with mean $\sigma_u^{-1}$. Hence, cooperativity and bursting have no effect on this switch. The switch from $G$ to $G^*$ occurs via $G + P \rightarrow G^*$, which is a nonlinear reaction with rate $\sigma_b$; in this case it is much more difficult to obtain the FPT because the process is nonlinear (most FPT theory is for linear reactions though there are exceptions[32]) and since there is a dependence on the instantaneous protein number that is affected by many different processes (transcription, degradation, bursting, cooperativity, etc). However, the LMA maps the above nonlinear reaction to a linear one, and thus enables us to obtain an approximate non-exponential expression for the FPT of switching from state $G$ to $G^*$ (given no protein initially in state $G$) for nonlinear GRNs (see "Methods" section). In Fig. 7a, we show the LMA's estimate of the FPT distribution for the feedback loop (Fig. 1a upper); the feedback loop with cooperativity, specifically two protein molecules binding the promoter in state $G$ (Fig. 2b) and the feedback loop with protein bursting and a mean burst size of two protein molecules (Fig. 2a) (the four parameters which are common to all three GRNs are fixed for comparison purposes; see Fig. 7 caption for details). The estimates are close to the FPT calculated using the SSA thus verifying the LMA's accuracy. The mean time to switch from $G$ to $G^*$ in a feedback loop is decreased considerably by cooperativity and slightly by bursting; this was observed for all parameter sets, which we studied. We also found out using the LMA that over a large region of parameter space, the mean first-passage time $\tau$ is approximately described by a simple power law in two parameters (Fig. 7b):

$$\tau \propto \sigma_b^{-3/5}\rho_u^{-4/5}, \quad \text{feedback loop with and without bursting} \quad (6)$$

$$\tau \propto \sigma_b^{-1/3}\rho_u^{-4/5} \quad \text{feedback loop with cooperativity} \quad (7)$$

For the case of the feedback loop with no cooperativity, it is possible to derive an exact solution for the FPT of switching from state $G$ to $G^*$ in steady-state conditions and hence this provides another means to evaluate the accuracy of the LMA (see Supplementary Note 4 for details). This comparison is shown in Fig. 7c, where we show that the error in the LMA's estimate of the FPT distribution (measured by the HD) increases with $\sigma_b$ (the rate parameter controlling the degree of nonlinearity in the GRN), in agreement with our previous error analysis for time-dependent protein distributions. Nevertheless, the high accuracy of the LMA for predicting first-passage time distributions is visually discernible in all cases.

Next we turn our attention to the sensitivity of nonlinear GRNs to noise. The coefficient of variation of protein number fluctuations defined as the ratio of the standard deviation of the fluctuations and the mean protein numbers is a common measure of the size of intrinsic noise. It is often the case that noise needs to be controlled such that the smooth performance of a certain cellular function is guaranteed[33]. The question then is: which parameter tweaking leads to the largest and smallest changes in

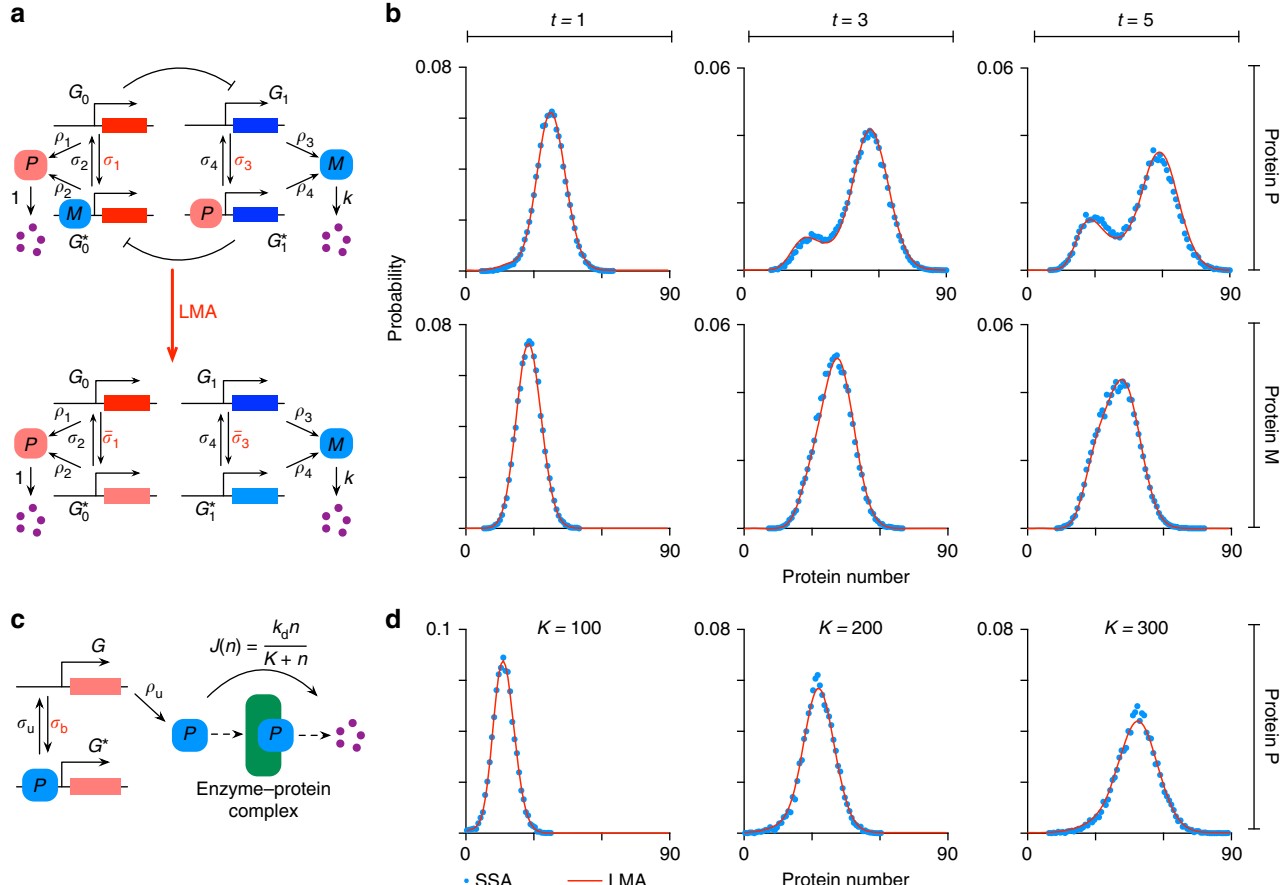

**Fig. 5** LMA for the toggle switch and for a feedback loop with nonlinear protein degradation. **a** (upper panel) Illustrates a toggle switch (with two active and two inactive promoters), whereby two proteins $P$ and $M$ are expressed and mutually repress each other. (lower panel) illustrates how the toggle switch system is decoupled by the LMA into two independent linear networks. **b** Shows the LMA predictions for time-evolution of the marginal distributions of the two proteins, $P$ and $M$, in the toggle switch for three different time points. The parameters are: $\rho_1 = 60$, $\rho_2 = 25$, $\rho_3 = 45$, $\rho_4 = 30$, $\sigma_1 = \sigma_3 = 0.004$, $\sigma_2 = \sigma_4 = 0.25$ and $k = 1$. **c** Shows the feedback loop with nonlinear protein degradation via an effective Michaelis–Menten reaction. **d** Shows the LMA predictions for steady-state protein distributions of the system shown in **c** for three different values of the Michaelin–Menten constant $K$. The other parameters are: $\rho_u = 3$, $\rho_b = 0$, $\sigma_b = 4 \times 10^{-4}$, $\sigma_u = 0.25$ and $k_d = 20$

the coefficient of variation squared in steady-state conditions? Whilst this is computationally very expensive to answer using the SSA over large areas of parameter space, with the LMA it can be addressed straightforwardly. We used the LMA to compute the logarithmic sensitivity[34] of the coefficient of variation squared to the four rate parameters common to all three non-oscillatory loops (namely $\rho_u$, $\rho_b$, $\sigma_b$ and $\sigma_u$) over a large swath of parameter space. The results are summarized in pie chart form in Fig. 8. For all feedback loops, independent of whether the protein was repressing or activating its own production, the most sensitive parameter was in the vast majority of cases $\rho_b$ while the least sensitive parameter was one of the other three parameters (typically was either $\sigma_u$ or $\sigma_b$). Hence in summary, control of the size of the protein fluctuations can be most efficiently obtained by tweaking gene expression in state $G^*$.

Finally, we study differences in the stochastic bifurcation diagrams of the three types of feedback loop. The LMA reveals that, for some parameter values, the system has a unimodal steady-state distribution, whereas for other values it has a bimodal distribution, i.e., the noise causes the system to switch between two distinct states. This phenomenon is referred to as noise-induced bistability (NIB) since the deterministic rate equations do not show bistability[35–37]. We explored how the region in parameter space where NIB is observed depends on

cooperativity, protein bursting, the type of loop (positive or negative feedback) and the existence of timescale separation. The results are summarized in Fig. 9. Each of the subfigures in Fig. 9 is generated by calculating the steady-state protein distribution over parameter space: the white then indicates a unimodal distribution while a shade of red indicates a bimodal distribution. The three shades of red indicate three different parameter sets as described in the figure caption where a lighter shade of red indicates a larger distance from detailed balance conditions. The calculation is done using the LMA and direct numerical integration of the master equation (as in ref. [7]) and differences between the two are shown in black. For negative feedback, the black regions show where numerical integration predicts unimodality, whereas the LMA (incorrectly) predicts bimodality whereas for positive feedback, the black regions show the opposite situation. In all cases, the black regions are very small thus showing the accuracy of the LMA in capturing NIB. The plots comparing positive (activator) and negative (repressor) feedback show a slight increase in the region of space where there is NIB when there is positive feedback. A comparison of the three shades of red shows that the major factor determining NIB is not the feedback type but rather the difference between the rates of protein production in the two promoter states, i.e., the distance from detailed balance. The larger the difference between the two rates, the larger is the region

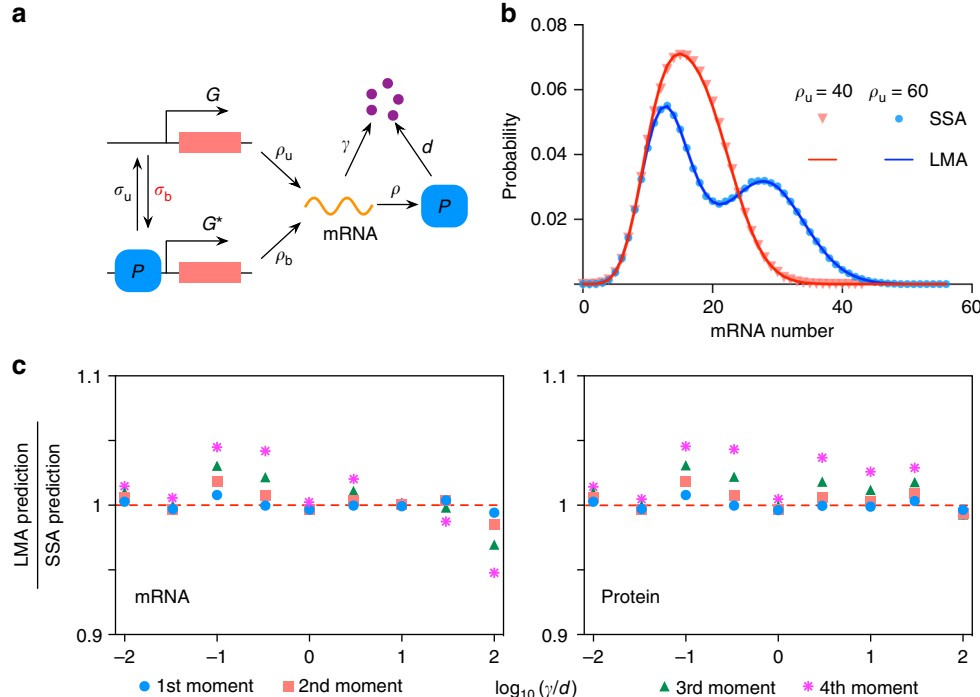

**Fig. 6** LMA for the feedback loop with explicit modeling of both mRNA transcription and protein translation. **a** Illustrates the nonlinear feedback loop. **b** Shows the LMA and SSA predictions for steady-state mRNA distribution (See Supplementary Note 7), showing that the LMA is able to capture the change in modality induced by a change in $\rho_u$. The rest of the parameters are: $\rho_b = 25$, $\sigma_b = 0.004$, $\sigma_u = 0.25$, $\rho = 3$, $\gamma = 2$ and $d = 1$. **c** Compares LMA and SSA predictions for the first to fourth moments (about zero) of both mRNA and protein numbers over a wide range of the mRNA-protein degradation ratio $\gamma/d$. The result indicates that the accuracy of the LMA is independent of time-scale separation assumptions. The parameters used are: $\rho_u = 5$, $\rho_b = 0$, $\rho = 100$, $\sigma_u = 1$, $\sigma_b = 1.5 \times 10^{-5}$. The nine pairs of ($\gamma$, $d$) are: (0.02, 2), (0.02, 0.6), (0.05, 0.5), (0.05, 0.15), (0.2, 0.2), (0.15, 0.05), (0.2, 0.02), (0.15, 0.005) and (0.2, 0.002), constituting a logarithmic span from −2 to 2. The ratio $\gamma/d$ varies over a range consistent with mammalian gene expression[31]

of parameter space where NIB is observed; for example for the negative feedback loop, the fraction of parameter space where NIB occurs is 59%, 36% and 5% for $(\rho_u, \rho_b) = (60, 25)$, (50, 25) and (40, 25), respectively (see Supplementary Table 1 for data of all cases shown in Fig. 9). We found that both cooperativity and bursting significantly reduce the size of this space and thus have an adverse effect on NIB. In Fig. 9b, we repeat the same exploration as in Fig. 9a, but using a method in the literature that assumes slow promoter switching, i.e., the timescale of promoter switching is much larger than the timescale of protein dynamics[12]. A comparison of Fig. 9a, b shows that the assumption of timescale separation tends to significantly over-estimate the size of parameter space where NIB exists, though capturing some of the major observed trends. The comparison also confirms that the LMA is free of underlying assumptions of timescale separation.

## Discussion

In this paper, we have introduced a new modeling framework, the LMA, based on a mapping of a nonlinear gene regulatory network to an approximately equivalent linear network. The approach rests on the following two assumptions: (i) a mean-field assumption and (ii) a time-averaging assumption. Specifically these two assumptions are needed to calculate the approximate time-dependent probability distribution solution of protein fluctuations but if one is interested in steady-state then only the first assumption is needed. The mean-field assumption essentially equates with assuming small protein fluctuations compared to the mean number of proteins when the promoter is unbound, a reasonable assumption given that protein numbers are typically

much larger than one. The time-averaging assumption implies that the probability distribution at time $T$ of a linear network with a time-dependent parameter $\alpha(t)$ is approximately given by solving the master equation assuming the parameter is a time-independent constant to obtain the solution at time $T$ and subsequently replacing the parameter (in the latter solution) by the time-averaged value of $\alpha(t)$ over the period $[0, T]$. This approximation was shown to correspond to the first term of the Magnus expansion of the time-dependent solution of the master equation.

We have verified that the LMA gives accurate probability distributions (compared to stochastic simulations and to direct numerical integration of the master equation) for feedback loops with and without cooperativity / bursting including those with time-dependent transcription. The accuracy was high independent of the type of feedback (positive or negative), of the modality of the distribution (unimodal or bimodal) and of the existence or lack of timescale separation. We found the accuracy of the LMA to be very high for short and long times and good at intermediate times. The likely reason is that to predict steady-state distributions the method needs only one assumption—the mean-field assumption whereas it needs in addition the time-averaging assumption for predictions in finite time (for short times the accuracy is necessarily high because of deterministic initial conditions). In all cases, the LMA well captures the changes in the shape of the distribution as a function of time, in particular, the transition from unimodal to bimodal behavior. The differences between the predicted and exact distribution are found to grow with the rate parameter controlling the nonlinear protein–promoter binding reaction and with the difference between the protein production rates of the two promoter states; however, these differences are typically barely noticeable to the

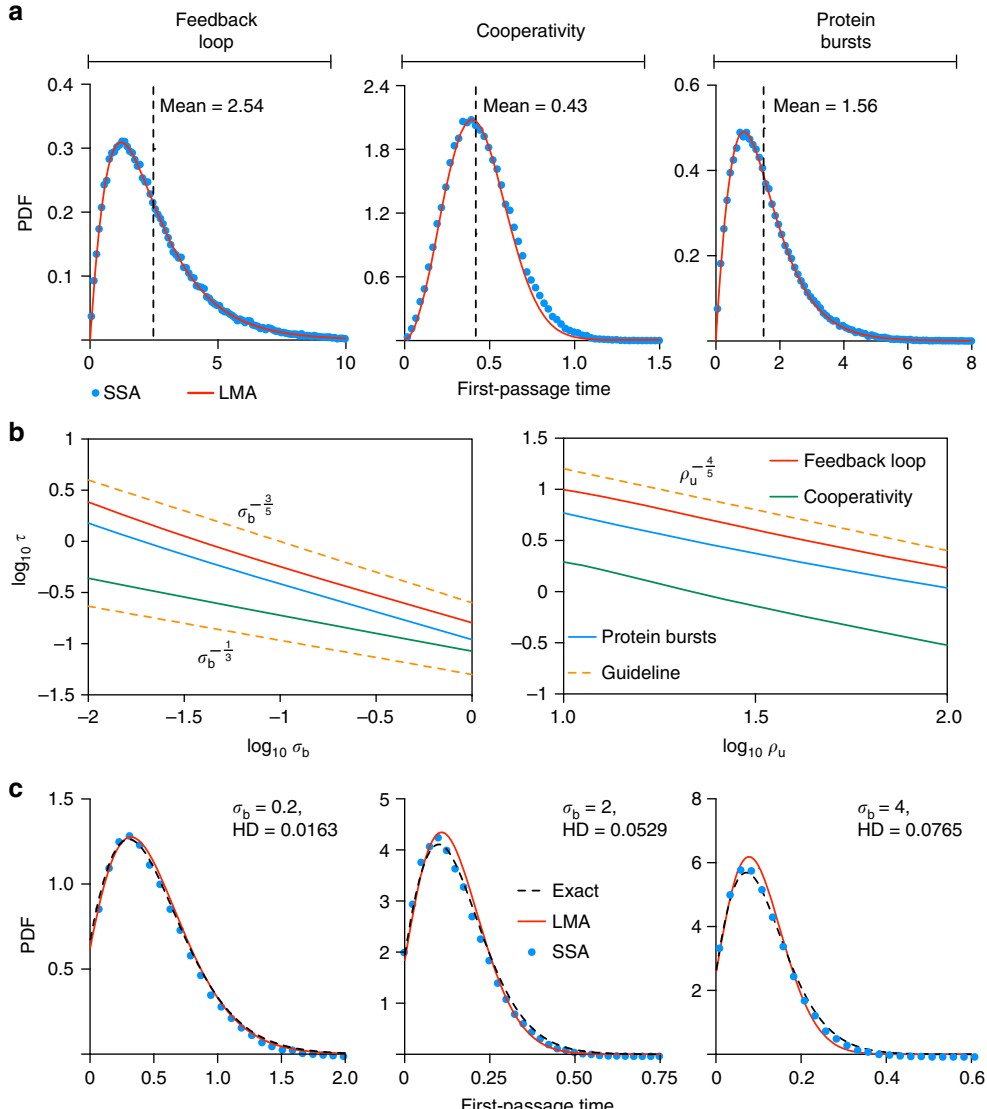

**Fig. 7** First-passage time (FPT) analysis of the switching from promoter state $G$ to $G^*$. **a** Compares the LMA prediction for the FPT probability density function (PDF) with zero proteins in state $G$ initially (red solid line) with that obtained using the SSA (blue dots) for the three non-oscillatory feedback loops. Cooperativity and protein bursts generally reduce the mean FPT. The parameter values are: $\rho_u = 60$, $\sigma_b = 0.01$, the cooperativity order $cp = 2$ and the mean protein bursts size $b = 2$. Note that there is no dependence of the FPT distribution on $\sigma_u$ and $\rho_b$ since the first-passage time process stops when the state changes to $G^*$. **b** Shows that the mean FPT ($\tau$) is a power law in two parameters $\sigma_b$ and $\rho_u$. The solid lines are the LMA predictions, while the orange dashed lines are guidelines indicating the power law. **c** Compares the LMA prediction for the FPT distribution in steady-state conditions (red solid line) with that obtained using the SSA (blue dots) and with the exact solution (Eq. (18) in Supplementary Note 4) for the feedback loop with no cooperativity as a function of $\sigma_b$

naked eye, except for some intermediate times. We also found that for nonlinear GRNs with an external input oscillatory signal, the LMA's predictions are accurate for input frequencies far from the intrinsic resonant frequency of the GRN itself; this is due to the filtering properties of the time-averaging assumption. We also used the LMA to study how cooperativity and protein bursting affect the first-passage time distribution governing promoter switching, the sensitivity of the coefficient of variation squared to parameter perturbation and the stochastic bifurcation diagram. The extensive study over large swaths of parameter space was made possible by the fact that the LMA provides closed-form solutions for the protein distributions. This is a distinct computational advantage over the stochastic simulation algorithm and also over the finite-state projection algorithm (see Supplementary Note 3 and Supplementary Fig. 2 for details of the comparison of CPU time of the various algorithms).

The LMA, of course, cannot possibly solve the master equations of all gene regulatory networks encountered in nature. In particular when the nonlinear GRN has also bimolecular reactions that are not of the protein–promoter type, the LMA mapping does not lead to a linear GRN though it is still a simpler GRN than the original one. In such a case, it is typically difficult to solve exactly the master equation of the simplified GRN. Likely, progress can then be made by replacing the bimolecular reactions (not involving protein and promoter) by an effective first-order reaction/s such that one has again an effective linear GRN. For example, for GRNs, for which the protein is catalyzed by an enzyme, the catalysis can be effectively modeled by a first-order decay reaction for the protein with a Michaelis–Menten rate (as shown in one of our examples). This additional linearization might not always be possible or else even if possible it might still lead to unsolvable or very difficult to solve master equations; this

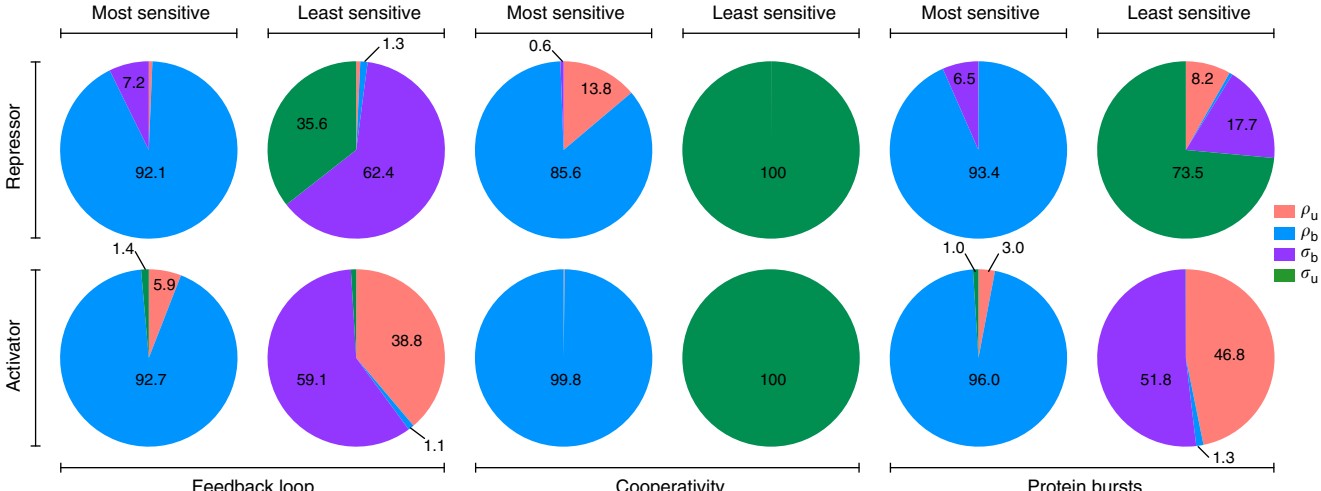

**Fig. 8** Sensitivity of the coefficient of variation squared to parameter perturbation in steady-state conditions. The pie charts show the most and least sensitive parameters for the three types of non-oscillatory feedback loops, in activating (positive feedback) or repressing mode (negative feedback). Out of the four variables, $\rho_b$ is the most sensitive parameter and occupies almost 90% in each of the six cases, whereas the least sensitive parameter is typically either $\sigma_b$ or $\sigma_u$. The activator stands for $\rho_b > \rho_u$, whereas the repressor means $\rho_u > \rho_b$. The sensitivity results are obtained by using the LMA to calculate the logarithmic sensitivity of the coefficient of variation squared to the four parameters ($\sigma_b$, $\sigma_u$, $\rho_b$ and $\rho_u$) on a regular lattice over the space: $\rho_b, \rho_u \in [1, 10^2]$ and $\sigma_b, \sigma_u \in [10^{-2}, 1]$. The lattice spacing is 5 for $\rho_b, \rho_u$ and 0.05 for $\sigma_b, \sigma_u$. The cooperativity order was cp = 2 for the feedback loop with cooperativity and the mean protein bursts size was $b = 2$ for the feedback loop with bursting

has to be ascertained on a case-by-case basis and no general statements can be made in this regard.

We finish by noting that the LMA has significant advantages over current methods in the literature. Unlike the linear-noise approximation, it does not assume the distribution is Gaussian and that the means are well described by the deterministic rate equations. It does not assume timescale separation, a common assumption in the literature. It is also superior to moment-closure methods[38–40] since there is no underlying assumption of a distribution of any kind and also since it does not just give the moments but also the distribution itself (a detailed study of the accuracy of the moments provided by the LMA and comparison with common moment-closure methods is under investigation). The LMA also provides distributions in analytical or semi-analytical form for all times, a clear advantage over methods based on distribution reconstruction using maximum entropy[23] or finite-state projection[21]. Hence concluding, the newly devised LMA method provides a new tool for the systematic exploration of the stochastic properties of nonlinear GRNs in systems and synthetic biology.

## Methods

**Master and moment equations.** Consider a chemical reaction network involving $N$ distinct species interacting with each other in a well-stirred volume $\Omega$ via a set of $R$ reactions $\sum_{i=1}^{N} s_{ir} X_i \xrightarrow{k_r} \sum_{i=1}^{N} p_{ir} X_i$, where $X_i$ stands for species $i$ ($i = 1, 2, \ldots, N$), $r = 1, 2, \ldots, R$ and the stoichiometric coefficients $s_{ir}$ and $p_{ir}$ are nonnegative integers specifying, the molecule numbers of reactants and products involved in reaction $r$, respectively. $k_r$ is the rate constant of reaction $r$. The associated CME can be written as:

$$\partial_t P(\mathbf{n}, t) = \sum_{r=1}^{R} f_r(\mathbf{n} - \mathbf{S}_r) P(\mathbf{n} - \mathbf{S}_r, t) - \sum_{r=1}^{R} f_r(\mathbf{n}) P(\mathbf{n}, t), \quad (8)$$

where $\mathbf{n} = [n_1, n_2, \ldots, n_N]^\top$ is the state vector of species molecule numbers, $P(\mathbf{n}, t)$ is the probability of the system being in state $\mathbf{n}$ at time $t$[4]. The i-th entry of the vector $\mathbf{S}_r$ is given by $p_{ir} - s_{ir}$, and $f_r(\mathbf{n})$ is the propensity function. The propensity function for the $r^{th}$ reaction assuming mass-action kinetics is then given by[19]:

$$f_r(\mathbf{n}) = k_r \Omega \prod_{i=1}^{N} \frac{n_i!}{(n_i - s_{ir})! \Omega^{s_{ir}}}. \quad (9)$$

Furthermore, we shall absorb powers of $\Omega$ in to $k_r$ so that the latter has units of inverse time for all reaction types.

The moment equations quantify the time evolution of moments. They can be derived directly from the CME, and have the compact form:

$$\partial_t \langle n_i \ldots n_l \rangle = \sum_{r=1}^{R} \langle (n_i + S_{ir}) \ldots (n_l + S_{lr}) f_r(\mathbf{n}) \rangle - \sum_{r=1}^{R} \langle n_i \ldots n_l f_r(\mathbf{n}) \rangle. \quad (10)$$

where $\langle n_i \ldots n_l \rangle = \sum_{n_i=0}^{\infty} \ldots \sum_{n_l=0}^{\infty} n_i \ldots n_l P(\mathbf{n}, t)$, the angled brackets denote the expectation operator and $S_{ij} = p_{ij} - s_{ij}$.

**LMA for feedback loops with and without cooperativity.** Here we provide details of the LMA for the nonlinear GRNs involving a feedback loop (Fig. 1a upper) and the feedback loop with cooperativity (Fig. 2b). For the latter, we shall here consider in detail the case of two proteins binding cooperatively to the promoter (cp = 2) and then briefly show how the procedure can be easily extended to the case of any number of proteins binding cooperatively.

Let the number of proteins, the unbound promoter state $G$ and the bound promoter state $G^*$ be denoted by $n_p$, $n_g = 1$ and $n_g = 0$, respectively. By the LMA, both types of nonlinear GRNs map onto the same linear GRN (Fig. 1a lower) whose stochastic dynamics is described by a master equation of the type given by Eq. (8). This is an equation for the time-evolution of $P(n_p, n_g, t)$. For convenience, since $n_g$ can be in only two states, we write $P_{1-n_g}(n_p, t) = P(n_p, n_g, t)$ meaning that $P_0(n_p, t)$ is the probability that the promoter is in state $G$ and there are $n_p$ proteins at time $t$ and $P_1(n_p, t)$ is the probability that the promoter is in state $G^*$ and there are $n_p$ proteins at time $t$. Thus, it follows that we can write the master equation as a set of two coupled master equations, one for each state:

$$\partial_t P_0(n_p) = \rho_u \left[ P_0(n_p - 1) - P_0(n_p) \right] + (n_p + 1) P_0(n_p + 1)$$
$$- n_p P_0(n_p) + \sigma_u P_1(n_p) - \bar{\sigma}_b P_0(n_p),$$
$$\partial_t P_1(n_p) = \rho_b \left[ P_1(n_p - 1) - P_1(n_p) \right] + (n_p + 1) P_1(n_p + 1)$$
$$- n_p P_1(n_p) - \sigma_u P_1(n_p) + \bar{\sigma}_b P_0(n_p). \quad (11)$$

Note that the argument $t$ is suppressed for notation simplicity. Note also that time and the parameters are dimensionless since we divided by the rate of protein degradation (as done in ref. [7]). In ref. [5], an exact time-dependent solution of the same master equations was obtained for the special case $\rho_u = 0$. It is straightforward to use the same method to treat the more general case of non-zero $\rho_u$, which leads to the exact solution for the probability distribution of the number of proteins at time $t$:

$$P(n_p, t) = \frac{1}{n_p!} \frac{d^{n_p}}{dw^{n_p}} (G_0(w, t) + G_1(w, t))|_{w=-1}, \quad (12)$$

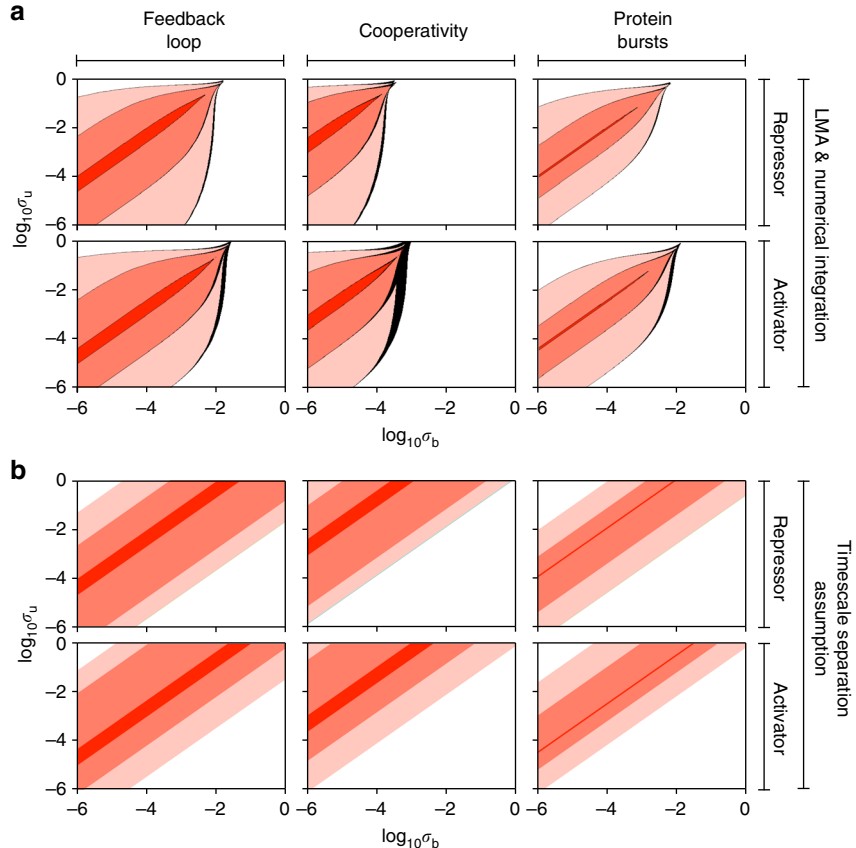

**Fig. 9** Stochastic bifurcation diagram for non-oscillatory feedback loops. **a** The red shaded areas denote the regions of parameter space where the steady-state distribution of protein numbers is bimodal, the white areas shows the regions where the distribution is unimodal and the black areas show regions where the LMA and direct numerical integration of the master equations disagree on the number of modes of the distribution. The black regions are small and thus verify the accuracy of the LMA. The shade of red indicates the difference between gene expression in the two promoter states, where a lighter shade indicates a larger difference. Specifically from inside to outside, the dark red corresponds to $\rho_u = 40$, $\rho_b = 25$ (repressor) and $\rho_u = 25$, $\rho_b = 40$ (activator), the medium red corresponds to $\rho_u = 50$, $\rho_b = 25$ (repressor) and $\rho_u = 25$, $\rho_b = 50$ (activator), and the light red corresponds to $\rho_u = 60$, $\rho_b = 25$ (repressor) and $\rho_u = 25$, $\rho_b = 60$ (activator). The cooperativity order was cp $= 2$ for the feedback loop with cooperativity and the mean protein bursts size was $b = 2$ for the feedback loop with bursting. Generally the region of parameter space where bimodality is present is decreased by cooperativity and by bursting but is almost unaffected by the type of feedback (activating or repressing). **b** This is the same as **a** except that the number of modes of the steady-state distribution of protein numbers is calculated using a method in the literature which assumes timescale separation, i.e. slow promoter switching[12]. While this method captures the salient features of the bifurcation diagrams in **a** it also significantly over-estimates the extent of bimodality and thus illustrates the advantage of the LMA over timescale separation methods

where,

$$G_0(w,t) = \exp(\rho_b w)\Big[f(we^{-t})(-\rho_\Delta w)^{1-\Sigma}M\big(1 - \bar{\sigma}_b, 2 - \Sigma, -\rho_\Delta w\big) \\ + g(we^{-t})M\big(1 + \sigma_u, \Sigma, -\rho_\Delta w\big)\Big], \quad (13)$$

$$G_1(w,t) = \sigma_u^{-1}\exp(\rho_b w)\Big[-\sigma_u f(we^{-t})(-\rho_\Delta w)^{1-\Sigma}M\big(-\bar{\sigma}_b, 2 - \Sigma, -\rho_\Delta w\big) \\ + \bar{\sigma}_b g(we^{-t})M\big(\sigma_u, \Sigma, -\rho_\Delta w\big)\Big], \quad (14)$$

where $w = z - 1$ and the generating functions are defined as $G_i(z,t) = \sum_{n_p=0}^{\infty} z^{n_p} P_i\big(n_p, t\big)$. The function $M(\cdot, \cdot, \cdot)$ represents the Kummer function and we have also used the definitions $\rho_\Delta = \rho_b - \rho_u$, $\Sigma = \sigma_u + \bar{\sigma}_b + 1$, $f(w) = \frac{\bar{\sigma}_b}{\Sigma - 1}(-\rho_\Delta w)^{\Sigma-1} e^{-\rho_u w} M(\sigma_u, \Sigma, -\rho_\Delta w)$ and $g(w) = \frac{\sigma_u}{\Sigma - 1} e^{-\rho_u w} M(-\bar{\sigma}_b, 2 - \Sigma, -\rho_\Delta w)$. We have assumed the initial conditions to be zero protein in state $G$ which translates in to the conditions: $P_0(0,0) = 1$, $P_0(n_p, 0) = 0$ for $n_p > 0$ and $P_1(n_p, 0) = 0$ for all $n_p$.

In steady-state conditions, the solution simplifies to:

$$P\big(n_p\big) = \frac{1}{n_p!}\frac{d^{n_p}}{dw^{n_p}}G(w)\big|_{w=-1}, \quad (15)$$

where

$$G(w) = \frac{\exp(\rho_b w)\sigma_u}{\sigma_u + \bar{\sigma}_b}M\big(1 + \sigma_u, \Sigma, -\rho_\Delta w\big) + \frac{\exp(\rho_b w)\bar{\sigma}_b}{\sigma_u + \bar{\sigma}_b}M\big(\sigma_u, \Sigma, -\rho_\Delta w\big). \quad (16)$$

Next we have to determine the value of the effective parameter $\bar{\sigma}_b$ using the LMA's mean-field assumption. From Eq. (10), one can obtain the moment equations of the linear GRN up to the third order:

$$\mathcal{M}_{FL}(\bar{\sigma}_b): \begin{cases} \partial_t \langle n_p \rangle = \rho_u \langle n_g \rangle + \rho_b\big(1 - \langle n_g \rangle\big) - \langle n_p \rangle, \\ \partial_t \langle n_g \rangle = -\bar{\sigma}_b \langle n_g \rangle + \sigma_u\big(1 - \langle n_g \rangle\big), \\ \partial_t \langle n_p^2 \rangle = 2(\rho_u - \rho_b)\langle n_p n_g \rangle + (2\rho_b + 1)\langle n_p \rangle - 2\langle n_p^2 \rangle + (\rho_u - \rho_b)\langle n_g \rangle + \rho_b, \\ \partial_t \langle n_p n_g \rangle = \rho_u \langle n_g \rangle + \sigma_u \langle n_p \rangle - (1 + \bar{\sigma}_b + \sigma_u)\langle n_p n_g \rangle, \\ \partial_t \langle n_p^2 n_g \rangle = (2\rho_u + 1)\langle n_p n_g \rangle - (\sigma_u + \bar{\sigma}_b + 2)\langle n_p^2 n_g \rangle + \rho_u \langle n_g \rangle + \sigma_u \langle n_p^2 \rangle. \end{cases} \quad (17)$$

Note that $\mathcal{M}_{FL}(\bar{\sigma}_b)$ is a set of closed ODEs given that $\bar{\sigma}_b$ is specified. The proposed LMA rests upon the idea of parametrizing $\bar{\sigma}_b$ by means of the conditional mean. Specifically, for the nonlinear feedback loop without cooperativity, since the nonlinear reaction is $P + G \rightarrow G^*$, then $\bar{\sigma}_b = \sigma_b \langle n_p | n_g = 1 \rangle = \sigma_b \langle n_p n_g \rangle / \langle n_g \rangle$, where we used the fact that $n_g$ is a Boolean variable. Similarly, for the nonlinear

feedback loop with cooperative order cp = 2, since the nonlinear reaction is $2P + G \to G^{\star}$ then $\bar{\sigma}_b = \sigma_b \left\langle n_p \left( n_p - 1 \right) | n_g = 1 \right\rangle = \sigma_b \left\langle n_p \left( n_p - 1 \right) n_g \right\rangle / \left\langle n_g \right\rangle$.

Subsequently, we solve the set of differential equations with initial conditions $\left\langle n_p \right\rangle = \left\langle n_p^2 \right\rangle = \left\langle n_p n_g \right\rangle = \left\langle n_p^2 n_g \right\rangle = 0$, $\left\langle n_g \right\rangle = 1$ and with the aforementioned $\bar{\sigma}_b$ parameterization, i.e., solving $\mathcal{M}_{FL}(\bar{\sigma}_b = \sigma_b \left\langle n_p n_g \right\rangle / \left\langle n_g \right\rangle)$ for feedback loop or $\mathcal{M}_{FL}\left( \bar{\sigma}_b = \sigma_b \left( \left\langle n_p^2 n_g \right\rangle - \left\langle n_p n_g \right\rangle \right) / \left\langle n_g \right\rangle \right)$ for cooperative network with cp = 2 on the time interval $[0, t]$. We denote the solved moments of interest at time $t'$ as $\left\langle n_g \right\rangle_{t'}$, $\left\langle n_p n_g \right\rangle_{t'}$ and $\left\langle n_p^2 n_g \right\rangle_{t'}$, where $0 \leq t' \leq t$. Hence, the effective time-dependent constants in the linear GRN are given by:

$$\bar{\sigma}_b(t') = \sigma_b \frac{\left\langle n_p n_g \right\rangle_{t'}}{\left\langle n_g \right\rangle_{t'}}, \; \bar{\sigma}_b(t') = \sigma_b \frac{\left\langle n_p^2 n_g \right\rangle_{t'} - \left\langle n_p n_g \right\rangle_{t'}}{\left\langle n_g \right\rangle_{t'}}, \quad (18)$$

for the noncooperative and cooperative feedback loops, respectively.

From these, we can compute the the time-averaged value of the effective parameter $\bar{\sigma}_b$ at time $t$:

$$\bar{\sigma}_b^* = \frac{\sigma_b}{t} \int_0^t \frac{\left\langle n_p n_g \right\rangle_{t'}}{\left\langle n_g \right\rangle_{t'}} \, dt' \; \text{and} \; \bar{\sigma}_b^{**} = \frac{\sigma_b}{t} \int_0^t \frac{\left\langle n_p^2 n_g \right\rangle_{t'} - \left\langle n_p n_g \right\rangle_{t'}}{\left\langle n_g \right\rangle_{t'}} \, dt',$$

for non-cooperative and cooperative loops, respectively. This is the time-averaging assumption of the LMA.

Finally the approximate probability distribution at time $t$ of the nonlinear GRN without cooperativity is given by Eqs. (12–14) with $\bar{\sigma}_b$ replaced by $\bar{\sigma}_b^*$ and for cooperativity the distribution is given by Eqs. (12–14) with $\bar{\sigma}_b$ replaced by $\bar{\sigma}_b^{**}$.

In steady-state, the solution is simpler. The moment equations can be solved with the time-derivative set to zero, i.e., $\mathcal{M}_{FL}\left( \bar{\sigma}_b = \sigma_b \left\langle n_p n_g \right\rangle / \left\langle n_g \right\rangle \right) = 0$, leading to explicit expressions for $\left\langle n_g \right\rangle$ and $\left\langle n_p n_g \right\rangle$ from which one can calculate the effective parameter $\bar{\sigma}_b^* = \sigma_b \left\langle n_p n_g \right\rangle / \left\langle n_g \right\rangle$ of the non-cooperative feedback loop:

$$\bar{\sigma}_b^* = \frac{-1 + \rho_b \sigma_b - \sigma_u + \sqrt{\left(1 - \rho_b \sigma_b + \sigma_u\right)^2 + 4\rho_u \sigma_b (1 + \sigma_u)}}{2}. \quad (19)$$

For the cooperative feedback loop, the effective parameter $\bar{\sigma}_b^{**} = \sigma_b \left( \left\langle n_p^2 n_g \right\rangle - \left\langle n_p n_g \right\rangle \right) / \left\langle n_g \right\rangle$ can be obtained in a similar way. It is found to be the solution of a third-order polynomial given by:

$$\bar{\sigma}_b^{**} = \sigma_b \frac{\rho_b^2 \bar{\sigma}_b^{**} \left(1 + \bar{\sigma}_b^{**}\right) + 2\rho_b \rho_u \bar{\sigma}_b^{**} (1 + \sigma_u) + \rho_u^2 (1 + \sigma_u)(2 + \sigma_u)}{\left(1 + \bar{\sigma}_b^{**} + \sigma_u\right)\left(2 + \bar{\sigma}_b^{**} + \sigma_u\right)}. \quad (20)$$

Finally, the approximate steady-state probability distribution of the nonlinear GRN without cooperativity is given by Eqs. (15) and (16) with $\bar{\sigma}_b$ replaced by $\bar{\sigma}_b^*$ in Eq. (19) and for cooperativity the distribution is given by Eqs. (15) and (16) with $\bar{\sigma}_b$ replaced by $\bar{\sigma}_b^{**}$ as obtained from solving Eq. (20). See the Supplementary Note 3 for details of an efficient numerical implementation of the LMA of the feedback loop using Mathematica.

The procedure can also be easily extended for the case of general number of proteins cp = n binding cooperatively to the promoter. In this case by the mean-field approximation, the effective parameter is:

$$\bar{\sigma}_b^{***} = \sigma_b \frac{\left\langle \prod_{i=0}^{n-1} \left( n_p - i \right) n_g \right\rangle}{\left\langle n_g \right\rangle}.$$

By substituting in the moment equations up to the order of $\left\langle n_p^n n_g \right\rangle$ and solving in steady-state, one finds that the effective rate constant is the solution of the following implicit function:

$$\bar{\sigma}_b^{***} = \sigma_b \frac{\sum_{i=0}^{n} C_i^n \rho_u^i \rho_b^{n-i} \prod_{j=1}^{i} (j + \sigma_u) \prod_{j=0}^{n-1-i} \left( j + \bar{\sigma}_b^{***} \right)}{\prod_{i=1}^{n} \left( \bar{\sigma}_b^{***} + \sigma_u + i \right)}.$$

Finally, the approximate steady-state probability distribution of the nonlinear GRN with cooperativity order n is given by Eqs. (15) and (16) with $\bar{\sigma}_b$ replaced by $\bar{\sigma}_b^{***}$ as obtained from solving the implicit equation above. The construction of the time-dependent solution parallels that previously shown for the special case of n = 2.

**LMA for feedback loop with protein bursts**. The feedback loop with protein bursting (Fig. 2a) is mapped onto the linear GRN described by:

$$G \xrightarrow{\rho_u} G + mP, \; G^{\star} \xrightarrow{\rho_b} G^{\star} + mP, \; G \xrightarrow{\bar{\sigma}_b} G^{\star}, \; G^{\star} \xrightarrow{\sigma_u} G, \; P \xrightarrow{1} \emptyset, \quad (21)$$

where $m$ is a discrete random variable sampled from the geometric distribution $\psi(m) = b^m/(1 + b)^{m+1}$ (see main text for justification of the choice of distribution). The mean burst size of gene expression is given by $b$. As in the previous example of noncooperative and cooperative feedback loops, we can derive coupled master equations of the linear GRN above:

$$\begin{aligned} \partial_t P_0\left( n_p \right) = &\rho_u \sum_{i=0}^{\infty} \psi(i) P_0\left( n_p - i \right) - \rho_u P_0\left( n_p \right) \\ &+ \left( n_p + 1 \right) P_0\left( n_p + 1 \right) - n_p P_0\left( n_p \right) \\ &+ \sigma_u P_1\left( n_p \right) - \bar{\sigma}_b P_0\left( n_p \right), \end{aligned} \quad (22)$$

$$\begin{aligned} \partial_t P_1\left( n_p \right) = &\rho_b \sum_{i=0}^{\infty} \psi(i) P_1\left( n_p - i \right) - \rho_b P_1\left( n_p \right) \\ &+ \left( n_p + 1 \right) P_1\left( n_p + 1 \right) - n_p P_1\left( n_p \right) \\ &- \sigma_u P_1\left( n_p \right) + \bar{\sigma}_b P_0\left( n_p \right). \end{aligned} \quad (23)$$

A time-dependent solution for these master equations is presently missing from the literature and we provide one in the Supplementary Note 1. Here we shall simply refer to the exact steady-state and time-dependent solutions as $\mathcal{S}_{PBSS}(\bar{\sigma}_b)$ and $\mathcal{S}_{PBTD}(\bar{\sigma}_b)$, respectively.

Using Eq. (10), one can obtain the corresponding moment equations up to the second order:

$$\mathcal{M}_{PB}(\bar{\sigma}_b): \begin{cases} \partial_t \left\langle n_p \right\rangle = \rho_u b \left\langle n_g \right\rangle + \rho_b b \left(1 - \left\langle n_g \right\rangle\right) - \left\langle n_p \right\rangle, \\ \partial_t \left\langle n_g \right\rangle = \sigma_u \left(1 - \left\langle n_g \right\rangle\right) - \bar{\sigma}_b \left\langle n_g \right\rangle, \\ \partial_t \left\langle n_p n_g \right\rangle = \rho_u b \left\langle n_g \right\rangle + \sigma_u \left\langle n_p \right\rangle - (1 + \bar{\sigma}_b + \sigma_u) \left\langle n_p n_g \right\rangle \end{cases}$$

Since the only nonlinear reaction in the nonlinear GRN is $P + G \to G^{\star}$ then by the LMA's mean-field assumption $\bar{\sigma}_b = \sigma_b \left\langle n_p | n_g = 1 \right\rangle = \sigma_b \left\langle n_p n_g \right\rangle / \left\langle n_g \right\rangle$. Solving the set of differential equations $\mathcal{M}_{PB}\left( \sigma_b \left\langle n_p | n_g = 1 \right\rangle \right)$ on the time interval $[0, t]$, we obtain the moments of interest at time $t'$ as $\left\langle n_g \right\rangle_{t'}$ and $\left\langle n_p n_g \right\rangle_{t'}$. The time-average $\bar{\sigma}_b^*$ is then constructed as before. The approximate solution for the probability distribution at time $t$ of the nonlinear GRN with protein bursts is given by $\mathcal{S}_{PBTD}(\bar{\sigma}_b^*)$.

For steady-state conditions, the moment equations can be solved explicitly (as before for the noncooperative and cooperative feedback loops) yielding:

$$\bar{\sigma}_b^* = \frac{1}{2}\left[ -1 + \rho_b b \sigma_b - \sigma_u + \sqrt{\left(1 - \rho_b b \sigma_b + \sigma_u\right)^2 + 4\rho_u b \sigma_b (1 + \sigma_u)} \right].$$

The approximate steady-state solution for the probability distribution of the nonlinear GRN with protein bursts is then given by $\mathcal{S}_{PBSS}(\bar{\sigma}_b^*)$.

**LMA for feedback loop with oscillatory transcription**. The feedback loop with oscillatory transcription (Fig. 2c) is mapped onto the same linear GRN used for cooperative and noncooperative feedback loops, namely that shown in Fig. 1a lower except for the parameters $\rho_u$ and $\rho_b$ which become $\rho_u \ell_t$ and $\rho_b \ell_t$, where $\ell_t = 1 + \text{Am} \cos(k\pi t)$ is an oscillatory function, where Am is the amplitude and $k$ is the frequency. Note that $0 < \text{Am} < 1$ such that the protein production rate in each promoter state is positive at all times. The master equations and moment equations are thus given by Eqs. (11) and (17), respectively, with the aforementioned substitutions. An explicit time-dependent probability distribution solution of the master equations can be found in the Supplementary Note 2. The approximate time-dependent distributions of the nonlinear GRN can then be obtained by the same LMA procedure as for the noncooperative feedback loop.

**The time-averaging assumption in the LMA**. The chemical master equation can be compactly written as:

$$\frac{d\mathbf{P}(t)}{dt} = \mathbf{A}_L(t)\mathbf{P}(t), \quad (24)$$

where $\mathbf{P}(t) = [P_0(t), P_1(t), \ldots]^{\top}$ and $P_1(t)$ is the probability that the system is in state $i$ at time $t$. Each entry of the transition matrix $\mathbf{A}_L(t)$ is defined by the propensity function governing the transition from one state to another. By means of the Magnus expansion of linear differential equations[41–43], the solution of the

master equation at time $t = T$ can be written as:

$$\mathbf{P}(T) = \exp(\mathbf{\Omega}(T))\mathbf{P}_0,$$

and $\mathbf{\Omega}(T) = \sum_{i=1}^{\infty} \mathbf{\Omega}_i(T)$. The first two terms of this expansion are:

$$\mathbf{\Omega}_1(T) = \int_0^T \mathbf{A}_L(t)dt, \qquad (25)$$

$$\mathbf{\Omega}_2(T) = \frac{1}{2}\int_0^T dt_1 \int_0^{t_1} dt_2 [\mathbf{A}_L(t_1), \mathbf{A}_L(t_2)]. \qquad (26)$$

The convergence of this expansion has been extensively studied (for a review of known results see Section 2.7 in ref. [42]). The time-averaging assumption corresponds to the first term of the Magnus expansion, i.e., truncating the expansion to include only $\mathbf{\Omega}_1(T)$. This is since this term is the same as if we had to first solve Eq. (24) assuming a time-independent transition matrix, leading to $\mathbf{P}(T) = \exp(\mathbf{A}_L T)\mathbf{P}_0$ and then replace the time-independent transition matrix $\mathbf{A}_L$ in this result by the time-averaged matrix $\int_0^T \mathbf{A}_L(t)dt/T$. This first term of the Magnus series of the master equation can be shown to give a well-defined probability vector and hence is physically meaningful to consider the expansion to this order only (see Supplementary Note 8 for a proof). Hence the approximation error of our time-averaging assumption is given by the rest of the terms in the expansion. What follows is an analysis of this error, in particular we prove that the error is small for all times in the limit of small protein–promoter binding rate, $\sigma_b$.

First of all, we note that for nonlinear GRNs with constant rates, the time dependence of $\mathbf{A}_L(t)$ with the LMA arises from the mapping to a linear GRN with a time-dependent protein–promoter binding rate $\bar{\sigma}_b(t)$ (for example see Eq. (18) for the case of feedback loops with and without cooperativity). We now look at the time-dependence of $\bar{\sigma}_b(t)$. Since there are zero proteins initially, $\bar{\sigma}_b(0) = 0$ while for long times $\bar{\sigma}_b(t)$ approaches a constant steady-state value determined by the steady-state values of the moments, e.g. the value of $\langle n_p n_g \rangle$ and $\langle n_g \rangle$ for the non-cooperative feedback loop. Furthermore the approach to steady-state occurs exponentially (see Supplementary Note 8 for a proof).

Since the time dependence of $\mathbf{A}_L(t)$ stems from $\bar{\sigma}_b(t)$, it also follows that $\mathbf{A}_L(t)$ converges to $\mathbf{A}_L(\infty)$ exponentially. This implies the following two statements. There exist some positive real numbers $C_1$ and $\delta_1$ such that

$$\|\mathbf{A}_L(t) - \mathbf{A}_L(\infty)\| \le C_1 e^{-\delta_1 t}$$

for any $t$ and the matrix $\mathbf{A}_L(t)$ can be expressed in terms of the steady-state matrix, i.e.,

$$\mathbf{A}_L(t) = \mathbf{A}_L(\infty) + \sigma_b \mathbf{D}(t),$$

where $\mathbf{D}(t)$ is a discrepancy matrix and $\|\mathbf{D}(t)\| \le C_2 e^{-\delta_2 t}$ for some positive real numbers $\delta_2$ and $C_2$. Note that all norms are matrix 2-norms. Thus, for the Lie bracket, we have:

$$\begin{aligned}
\|[\mathbf{A}_L(t_1), \mathbf{A}_L(t_2)]\| = &\ \|[\mathbf{A}_L(\infty) + \sigma_b \mathbf{D}(t_1), \mathbf{A}_L(\infty) + \sigma_b \mathbf{D}(t_2)]\| \\
\le &\ \sigma_b \|\mathbf{A}_L(\infty)\|\|\mathbf{D}(t_1)\| \\
&\ + \sigma_b \|\mathbf{A}_L(\infty)\|\|\mathbf{D}(t_2)\| \\
&\ + \sigma_b^2 \|\mathbf{D}(t_1)\|\|\mathbf{D}(t_2)\| \\
\le &\ \sigma_b C_3 e^{-\delta_2 t_2}
\end{aligned}$$

for some positive real number $C_3$ and for any $t_2 \le t_1$. Therefore, the matrix norm $\mathbf{\Omega}_2(T)$ is upper bounded by:

$$\|\mathbf{\Omega}_2(T)\| \le \sigma_b \frac{C_3}{2}\int_0^T \int_0^{t_1} e^{-\delta_2 t_2}dt_2 = \frac{\sigma_b C_3}{2\delta_2^2}\left(\delta_2 T + e^{-\delta_2 T} - 1\right) \sim \sigma_b \mathcal{O}(T),$$

On the other hand, it is known that:

$$\lim_{T \to \infty} \frac{\|\mathbf{\Omega}_1(T)\|}{T} = \|\mathbf{A}_L(\infty)\|.$$

Thus, we have:

$$\lim_{T \to \infty} \frac{\|\mathbf{\Omega}_2(T)\|}{\|\mathbf{\Omega}_1(T)\|} \sim \sigma_b \mathcal{O}(1).$$

This result indicates that the approximation error is bounded in time and first order in $\sigma_b$. Applying similar arguments, it can be shown that all higher-order terms in the Magnus expansion ($\mathbf{\Omega}_i$, where $i \ge 3$) are bounded in time and have higher orders in $\sigma_b$ than the first two terms. In other words, one can conclude that the time-averaging assumption of the LMA is uniformly valid in time and accurate provided the protein–promoter binding rate $\sigma_b$ is small.

**The first-passage time distribution of promoter switching.** For our purposes, this is the distribution of the time it takes for the promoter to switch from $G$ to $G^*$, given there are $n$ proteins initially. We shall show this for the cooperative and noncooperative feedback loops (similar can be done for the bursty loop). Given the process, the only relevant reactions for this calculation are:

$$G \xrightarrow{\rho_u} G + P, \text{cp } P + G \xrightarrow{\sigma_b} G^*, P \xrightarrow{1} \emptyset, \qquad (27)$$

where cp = 1 for the noncooperative feedback loop, and cp > 1 for the cooperative feedback loop. The LMA maps this onto the simpler linear GRN:

$$G \xrightarrow{\rho_u} G + P, \ G \xrightarrow{\bar{\sigma}_b} G^*, P \xrightarrow{1} \emptyset \qquad (28)$$

Using the same recipe as before, one writes the moment equations, applies the mean-field assumption, obtains the relevant moments and then calculates the effective time-dependent constants $\bar{\sigma}_b(t)$. This implies the solution of the moment equations Eq. (17) with the constants $\rho_b$ and $\sigma_u$ set to zero (since the associated reactions are irrelevant to the first-passage time process as described above) and initial conditions $\langle n_P \rangle = n$, $\langle n_P^2 \rangle = n^2$, $\langle n_P n_g \rangle = n$, $\langle n_P^2 n_g \rangle = n^2$, $\langle n_g \rangle = 1$.

The first-passage time distribution to switch from $G$ to $G^*$ is given by $P(t_{FP} = t) = -\partial_t P_0(t)$[44], where $P_0(t)$ is the probability that the system is still in state $G$ at time $t$ given that initially it is in this state. Since the LMA maps the second-order reaction onto the linear reaction $G \to G^*$ with effective rate $\bar{\sigma}_b(t)$, it follows from elementary probability arguments that:

$$P_0(t) = \exp\left(-\int_0^t \bar{\sigma}_b(t')dt'\right).$$

Hence, the final expression for the first-passage time distribution in the LMA is given by:

$$P(t_{FP} = t) = \bar{\sigma}_b(t)\exp\left(-\int_0^t \bar{\sigma}_b(t')dt'\right).$$

**Code availability.** The Mathematica code solving the LMA for the simple nonlinear feedback loop (schematically shown in Fig. 1a (upper)) can be found at https://github.com/edwardcao3026/Linear-mapping-approximation. The details are provided in Supplementary Note 3.

**Data availability.** All relevant data are available from the authors.

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

## Acknowledgements

This work was supported by a grant from the BBSRC (BB/M018040/1).

## Author contributions

Z.C. and R.G. designed research, carried out research and wrote the manuscript.

## Additional information

**Competing interests:** The authors declare no competing interests.

