## [Peer Review File · Nature Communications]

Reviewers' comments:

Reviewer #1 (Remarks to the Author):

Review: **Linear mapping approximation of nonlinear gene regulatory networks**

Zhixing Cao and Ramon Grima

Overview: This paper presents a novel approach for analyzing stochastic models of nonlinear Gene Regulatory Networks (GRNs). The main difficulty in studying these models is that the Chemical Master Equation (CME) associated with the stochastic dynamics of these models is analytically intractable and simulation-based or numerical approaches are inadequate for systematic analyses of these models over large parameter spaces. The aim of this paper is to circumvent this problem by mapping the nonlinear GRN into a somewhat equivalent linear GRN whose CME is known to have an analytical solution. This approach is called the *Linear Mapping Approximation* (LMA) and it is shown to be very accurate in capturing the transient and steady-state solution to the CME for various types of nonlinear GRNs with features such as cooperativity, protein bursting and time-varying transcription rates. LMA does not require restrictive assumptions such as Gaussian distributions or time-scale separation, unlike previous approaches to study nonlinear GRNs in the stochastic setting. This generality is a big advantage for LMA and it is shown how LMA can accurately predict a whole range of important quantities/behaviors, such as effects of cooperativity and protein-bursting, transition to bimodality with time, stochastic bifurcation diagrams and first-passage time distributions for promoter switches.

Recommendation: The paper addresses a very important topic which would be of enormous interest to systems and synthetic biologists. However I find the approach of this paper mathematically unsatisfactory and limited in applicability. Therefore I cannot recommend publication of this paper in a journal like *Nature Communications*. My reasons for this negative decision are mentioned below in greater detail.

1. My main criticism of LMA is that there are several issues that limit its applicability beyond the simple models considered in the paper. This approach demands an analytical solution to the CME for the linear network which is a very restrictive assumption. It automatically precludes using nonlinear GRNs where kinetics is not of simple mass-action type but has some reactions with propensities in the form of nonlinear Hill functions. For many GRNs, the transcriptional feedback by protein is often modeled with a Hill function to capture the saturation effect. This is especially true for models with cooperativity. Another problem with LMA is that the mRNA dynamics is not considered in any of the considered examples. Even though mRNA might be short-lived, due to nonlinearity the delays caused by mRNA transcription step can get amplified rendering the model without mRNA completely inaccurate. The proposed approach will not work in many cases if mRNA activity is considered which would be problematic if one tries to use LMA to fit experimental data.

2. At the heart of LMA, is a moment matching procedure that allows one to obtain $\bar{\sigma}_b$ as a function of time t (Eq. 3 in the paper). Here the “moment-closure” is provided by the *local decorrelation* assumption which is used to obtain the linear network. How can LMA then be claimed to be conceptually superior to other moment-closure schemes, some of which are also doing moment-matching under various assumptions? Also there is circularity of logic here - from a nonlinear GRN first a linear GRN is obtained, and then moment equations for this linear GRN determine the key parameter $\bar{\sigma}_b$ which is used to derive the linear model. This approach is flawed, unless moments of nonlinear and linear versions of GRN match well, but this cannot be assumed at the start.
3. The authors claim that LMA provides “closed-form” solutions at all times, but this is inaccurate as numerical integration of the moment equations is required to obtain $\bar{\sigma}_b$ and its time-averaged value. Given this, I don’t see why LMA is superior to the FSP approach which not only computes the distribution numerically, but it also bounds the error explicitly.
4. Using time-averaging to obtain time-dependent solution is more of a hack than Science. Its mathematical justification in Section 4.5, based on linear first-order ODE is not very convincing.

Reviewer #2 (Remarks to the Author):

The manuscript by Cao and Grima presents a novel approximation method for obtaining the time-dependent probability distribution of gene regulatory networks. The idea behind the paper is novel to the best of my knowledge and is interesting. It is based on a local decorrelation assumption and a time averaging assumption to map the nonlinear GRNs to an appropriate linear GRN that is analytically solvable. The authors use their method to obtain analytical results for feedback systems with and without cooperativity and burstiness. They also use their method in several applications including an stochastic bifurcation and sensitivity analysis of the GRNs studies in the paper. This is a nice contribution to the literature and potentially suitable for publication in Nature Communication. But, the authors should address the following comments:

- The mapping the to the linear GRN the authors introduce is relatively general, however the examples they discuss are related to GRNs involving a single gene and ignoring mRNA. For this model relying on the results in reference 5 the authors have analytical solution that has been used throughout the paper. Could the approach be generalised to more complex GRNs, for example where mRNA is explicitly modelled, where promotor has more than 2 states, or most importantly the where the GRN includes more than one gene (e.g. the toggle switch or represilator). If so, this should be at least discussed or even better an example included. If not, this limitation should be clearly stated

2. Following on from first comment, I believe the mapping to the linear GRN can be done for more complex nonlinear GRNs, but the corresponding linear GRN may not be generally analytically solvable. In these cases, can the authors think of other advantage for this mapping in terms of analysis.

3. An analytical solution is useful in providing novel insights and in making the computation faster. Is there any new insight obtained from the analytical solutions obtained that could not be achieved with numerical simulation? As the authors state in the supplement, numerically evaluating the analytical solutions are complex and computationally intensive. The authors should include a figure that compares the computational advantage over SSA.

4. A short explanation on why Hellinger distance and not other measures of distance between distributions is used in the results would be useful.

Reviewer #3 (Remarks to the Author):

In this manuscript, the authors develop a novel approximation scheme that maps a specific class on nonlinear gene regulatory networks (GRNs) to linear GRNs to obtain explicit analytical distributions. The nonlinearity in the GRNs considered arises from feedback reactions involving protein binding to the promoter. The approximation scheme involves two assumptions: 1) a 'mean-field' assumption which involves replacing a fluctuating rate by its mean value (conditional on promoter state) and 2) a time-averaging assumption to obtain time-dependent probability distributions. The approximations are well-motivated and clearly explained and the corresponding analytical results are in good agreement with results from stochastic simulations. The authors apply their approach to a range of different GRNs and in the process also obtain useful new results for time-dependent problems in such GRNs. The approach developed can be generalized and applied to characterize steady-state properties of a more general class of nonlinear GRNs and should be of broad interest to the field. That said, the following points should be addressed to improve the manuscript:

1) The notation in section 4.2 (Methods) needs improvement. The authors denote the unbound promoter (G) by $n_g = 1$ and the bound promoter (G*) by $n_g = 0$. Then they write down the master equation for time evolution of $P(n_p, n_g)$ but the notation is confusing. In particular, they set $P_0(n_p) = P(n_p, n_g=1)$ and $P_1(n_p) = P(n_p, n_g=0)$ which can be confusing and should fixed.

2) Typically, generating functions are defined by $G(z) = \sum z^n P(n)$ whereas the authors choose $G(z) = \sum (z+1)^n P(n)$. They should consider adding a line further explaining/highlighting the choice so that readers wishing to compare results from different papers are aware of the different choices made in defining the generating function.

3) The discussion after Eqn (2) is slightly misleading. The authors point out that for the reversible reaction the protein number changes by ± 1 , whereas for the approximated linear case it does not change. This is correct but it is not the main source of error in the approximation in my opinion. For large mean protein numbers, the main source of error in the approximation is that they replace a fluctuating rate of transition by an average rate; the fact that the protein number changes by ± 1 in the two reactions is less likely to be significant. I think a calculation of the FPT for transition between promoter states (see point below) can also address this point further.

4) In the Applications of the LMA..(and in Section 4.6), the authors calculate the FPT for a switch between promoter states using their approximation scheme. However, this is done with the assumption that the initial state has zero proteins which is very restrictive and also not well justified. If we are in the steady-state limit, a more reasonable assumption is that the initial protein level is drawn from the steady-state protein distribution (or perhaps from the protein

distribution conditional on promoter state) rather than setting it to zero.

Furthermore, given the initial protein level, it should be possible (at least for the non-cooperative case) to calculate the FPT distribution exactly for Eqn. (27) (Section 4.6). The authors can simply write down the master equation for $P_{\{0\}}(n,t)$ (where n refers to the number of proteins at time t)

and obtain the FPT distribution $P(t_{\{FP\}} = t)$ by differentiating wrt time (as they have on pg 12) and summing over all n . Now it should be possible to solve the master equation for $P_{\{0\}}(n,t)$ using the method of characteristics. It would be instructive if the authors can compare this exact solution with the solution given by the LMA approximation (pg 13). In particular, it would be interesting to note if there is a correlation between errors in the LMA approximation for the steady-state or time-dependent distributions with the corresponding errors in the FPT distribution.

5) At different points in the text, the authors state that the LMA is exact when the condition of detailed balance is met. While this statement is correct, I feel it is somewhat misleading. For the cases considered, the feedback is irrelevant when detailed balance is satisfied, because the protein production rate is ρ regardless of the promoter state.

So if, instead of using the promoter switching rate obtained using the LMA, the authors used any arbitrary switching rate in the linear model, they would still obtain the exact steady-state distribution (a Poisson distribution with mean ρ). So it seems somewhat misleading to attribute this to the LMA and somewhat pointless to have a subsection in the supplementary material highlighting how the LMA result reduces to the exact result in this case.

6) While the authors have looked at different models, the range seems to be restricted to models with only 2 promoter states. This is likely because analytical solutions for the time-dependent problem may not be tractable in the general case with more than 2 promoter states. However, the steady-state distributions are also of broad interest and, as noted by the authors, their approach is more straightforward and involves less assumptions if we focus on steady-state alone. So it would be useful if the authors could comment on the applications of their approximation scheme to models with multiple (> 2) promoter states.

Response to referee comments for NCOMMS-18-00661

We thank all three reviewers for their constructive comments. We have thoroughly revised the manuscript to address all their concerns and suggestions. All changes in the manuscript are highlighted in red.

Reviewer #1

Reviewer Comment:

My main criticism of LMA is that there are several issues that limit its applicability beyond the simple models considered in the paper. This approach demands an analytical solution to the CME for the linear network which is a very restrictive assumption. It automatically precludes using nonlinear GRNs where kinetics is not of simple mass-action type but has some reactions with propensities in the form of non-linear Hill functions. For many GRNs, the transcriptional feedback by protein is often modeled with a Hill function to capture the saturation effect. This is especially true for models with cooperativity. Another problem with LMA is that the mRNA dynamics is not considered in any of the considered examples. Even though mRNA might be short-lived, due to nonlinearity the delays caused by mRNA transcription step can get amplified rendering the model without mRNA completely inaccurate. The proposed approach will not work in many cases if mRNA activity is considered which would be problematic if one tries to use LMA to fit experimental data.

Response:

We thank the reviewer for these detailed comments. Actually we already effectively had both Hill-type kinetics for transcriptional activation/repression and mRNA dynamics in our models, albeit they were implicit not explicit and we did not previously discuss this.

We have added a comment in Section 2.2 to explain this: "Note that an implicit description of mRNA exists in the model with protein bursting because protein burst sizes distributed according to a geometric distribution are obtained when the protein is produced by a fast intermediate mRNA, a common scenario in bacteria and yeast. Note also that while all the three systems are composed of reactions with mass-action propensities, in certain limits they reduce to systems composed of effective reactions with non-mass action propensities e.g. under quasi-equilibrium conditions between promoter and protein, the model of a feedback loop with cooperative binding reduces to an effective model describing protein production with a Hill-type propensity"

In other words, our model in terms of mass-action propensities and a model where the master equation has a propensity using Hill functions will agree in the quasi-equilibrium limit (as has been shown by several studies) and thus Hill kinetics are simply a special case of our model (this can also be inferred by imposing the quasi-equilibrium assumption between promoter and protein on the rate equations of

our model which leads to a single rate equation for the protein with a production rate given by Hill-type kinetics).

We have also extended our treatment of the LMA for the feedback loop with cooperativity in Methods Section 4.2 to describe the case of general “cp”, i.e, the number of proteins bound to the promoter (previously it was only for cp =2); hence in the quasi-equilibrium limit the LMA describes a system with general Hill coefficient “cp” describing the transcriptional activation / repression.

In addition, we have now studied a feedback loop with explicit mRNA and protein dynamics and also one where the protein degradation propensity is of the Michaelis-Menten type rather than the usual mass-action type. In both cases the LMA gives accurate results. A discussion of these systems can be found at the end of Section 2.2, the associated graphs in Figs 5 and 6 and details of the LMA for these circuits can be found in the SI Sections 5 and 7. We further note that the case that the referee mentions, i.e. the case where mRNA decay is short-lived, is completely within the scope of the LMA because the steady-state protein and mRNA distributions are known in this case for the linear network generated by the LMA. This is now discussed at the end of Section 2.2.

Reviewer Comment:

At the heart of LMA, is a moment matching procedure that allows one to obtain σ_b as a function of time t (Eq. 3 in the paper). Here the “moment-closure” is provided by the local decorrelation assumption which is used to obtain the linear network. How can LMA then claimed to be conceptually superior to other moment-closure schemes, some of which are also doing moment-matching under various assumptions? Also there is circularity of logic here - from a nonlinear GRN first a linear GRN is obtained, and then moment equations for this linear GRN determine the key parameter σ_b which is used to derive the linear model. This approach is flawed, unless moments of nonlinear and linear versions of GRN match well, but this cannot be assumed at the start.

Response:

We believe it is not correct to see the LMA as a type of moment-closure obtained through some type of moment matching procedure. We have revised the description of the “local decorrelation assumption” (now renamed as the mean-field assumption due to a suggestion by another reviewer) to clarify this. In particular the only assumption which the LMA does to go from a nonlinear to a linear model is to replace the effective stochastic rate to jump from promoter state G to G^ by its expectation. Everything follows logically from this one step: the mean stochastic rate is then exactly equal to Eq. 2 by the fact that n_g is a Boolean variable and the moment equations with this average rate are automatically closed and can be solved because the system is linear. The reason we claim that the LMA is superior to moment-closure methods is simply because invariably such methods lead only to approximations for the moments but not to the probability distribution (unless they are combined with some other method like max entropy which leads to even more assumptions and non-analytical distributions). In contrast, the LMA*

leads to distributions in many cases and where not, it still gives all the moments immediately (all orders are available because of the linear system obtained through the mapping) – this is now clarified by a new paragraph at the end of Section 2.1.

Reviewer Comment:

The authors claim that LMA provides “closed-form” solutions at all times, but this is inaccurate as numerical integration of the moment equations is required to obtain σ_b and its time-averaged value. Given this, I don’t see why LMA is superior to the FSP approach which not only computes the distribution numerically, but it also bounds the error explicitly.

Response:

We have updated the text to be more careful about our claims. Specifically in the paragraph after Eq. 5 and in the abstract we clarify that the LMA gives rise to either analytic or semi-analytic expressions for the distributions: “... for a nonlinear GRN with N protein-promoter binding reactions, the approximate time-dependent probability distribution given by the LMA is a closed-form distribution with N effective parameters to be determined numerically.” However note that this is for the time-dependent case. For the steady-state case, as we have shown, the distribution solutions of the LMA can be fully analytic.

The comment about why then not use the FSP is an interesting one and so we performed a case study to see the computational efficiency of the LMA vs FSP. This is reported in detail in the SI Section 3 and the associated SI Fig 2. We find that for all cases studied, the LMA significantly outperforms the FSP in computational time while retaining the same accuracy in the distribution – this is particularly so when the protein numbers are larger than a hundred (see Fig 2B) a common case in many GRNs, particularly eukaryotic ones. The reason for this is simply that the evaluation of the well-known functions in the closed-form solution (hypergeometric) and the numerical integral associated with the time-average can be computed very quickly by standard packages (we also found a way to bypass the previous computational limitation of having to take derivatives to obtain the distribution from the generating function and this has immensely speeded up the LMA evaluation down to fractions of a second in many cases; see SI Section 3 for details).

Regarding errors in the LMA; we have shown already that they stem almost exclusively from the time-averaging assumption (as the HD is practically zero in all cases in steady-state as shown in Figs 1 and 3). Our new rigorous derivation of this assumption in the Methods Section 4.5 gives an expression for the leading-order error made using this approximation. Hence the error can be estimated rigorously though this exercise is left for another paper as it is a considerable technical work in itself and our paper is already brimming with new results; the approach we have taken in the current paper is to estimate the error computationally as shown in Figs 1, 2 and 3.

Reviewer Comment:

Using time-averaging to obtain time-dependent solution is more of a hack than Science. Its mathematical justification in Section 4.5, based on linear first-order ODE is not very convincing.

Response:

We agree that previously this was the least convincing part. Encouraged by the referee's comments, we have now found a rigorous derivation of this assumption using the Baker-Campbell-Hausdorff series expansion. This is reported in the Methods Section 4.5. The time-averaging assumption comes out as the first term in the aforementioned series and the leading-order error is then given by the next term in the series.

Reviewer #2

Reviewer Comment:

The mapping the to the linear GRN the authors introduce is relatively general, however the examples they discuss are related to GRNs involving a single gene and ignoring mRNA. For this model relying on the results in reference 5 the authors have analytical solution that has been used throughout the paper. Could the approach be generalised to more complex GRNs, for example where mRNA is explicitly modelled, where promoter has more than 2 states, or most importantly the where the GRN includes more than one gene (e.g. the toggle switch or represilator). If so, this should be at least discussed or even better an example included. If not, this limitation should be clearly stated

Response:

We have now extended our LMA study to include feedback loops with explicit mRNA and protein dynamics, with nonlinear protein degradation propensity of the Michaelis-Menten type and as well to the multi-promoter toggle switch. In all cases the LMA gives accurate results. A discussion of these systems can be found at the end of Section 2.2, the associated graphs in Figs 5 and 6 and details of the LMA for these circuits can be found in the SI Sections 5, 6 and 7.

Reviewer Comment:

Following on from first comment, I believe the mapping to the linear GRN can be done for more complex nonlinear GRNs, but the corresponding linear GRN may not be generally analytically solvable. In these cases, can the authors think of other advantage for this mapping in terms of analysis.

Response:

We have added a comment in the discussion after Eq. 5 to clarify this particular case: “Note that independent of whether we are interested in the time-dependent or steady-state problem, when a closed-form solution for the linear GRN does not exist, the method still gives approximate expressions for all the moments of the nonlinear GRN using steps (i) and (ii); in this case its output is similar to moment-closure methods but with the advantage that we have made no implicit assumption on the form of the probability distribution solution of the chemical master equation.”

Reviewer Comment:

An analytical solution is useful in providing novel insights and in making the computation faster. Is there any new insight obtained from the analytical solutions obtained that could not be achieved with numerical simulation? As the authors state in the supplement, numerically evaluating the analytical solutions are complex and computationally intensive. The authors should include a figure that compares the computational advantage over SSA.

Response:

Closed-form expressions can be useful to see the dependencies of the distribution and its moments on the parameters. It is because of this that we can compute the parameter sensitivity and stochastic bifurcation diagrams over huge swaths of parameter space in short time (as shown in Fig. 8 and 9). In the previous version of the manuscript, we were evaluating the distribution by taking derivatives over the generating function. This led to numerical instabilities and computation times of the order of a min in some cases. Since then we have devised a method which completely avoids the use of derivatives and this has reduced the computation time to under a second in most cases and to few seconds for systems with large numbers of proteins. We have also carried out a case study comparing the CPU time of the LMA with the Finite State Projection Method (FSP) and with the SSA. These are discussed in the SI Section 3 and the associated SI Fig 2. Overall we find that typically the SSA takes tens of mins, FSP takes few mins while the LMA takes few seconds to produce distributions of comparable accuracy. We also note that because we have an analytical solution, the LMA is useful as a means to perform inference of parameters from experimental data, a topic that we are currently investigating.

Reviewer Comment:

A short explanation on why Hellinger distance and not other measures of distance between distributions is used in the results would be useful.

Response:

We have added a sentence on the Hellinger distance (HD) at the beginning of Section 2.2: “Note that the HD has the properties of being symmetric and satisfies

the triangle inequality thus implying that it is a distance metric on the space of probability distributions (unlike for example the commonly used Kullback-Leibler divergence)”.

Reviewer #3

Reviewer Comment:

The notation in section 4.2 (Methods) needs improvement. The authors denote the unbound promoter (G) by $n_g = 1$ and the bound promoter (G*) by $n_g = 0$. Then they write down the master equation for time evolution of $P(n_p, n_g)$ but the notation is confusing. In particular, they set $P_0(n_p) = P(n_p, n_g=1)$ and $P_1(n_p) = P(n_p, n_g=0)$ which can be confusing and should be fixed.

Response:

We agree with this comment. We have now defined at the beginning of Methods Section 4.2 that: $P_{\{1-n_g\}}(n_p, t) = P(n_p, n_g, t)$. This is the simplest way to make all our notation consistent, without having to rewrite all of the equations in the manuscript and SI.

Reviewer Comment:

Typically, generating functions are defined by $G(z) = \sum z^n P(n)$ whereas the authors choose $G(z) = \sum (z+1)^n P(n)$. They should consider adding a line further explaining/highlighting the choice so that readers wishing to compare results from different papers are aware of the different choices made in defining the generating function.

Response:

We have now changed this so that the definition of the generating function we use is the same as the conventional one and then we clarify that we express formulae in terms of $w = z - 1$. See after Eq. 14. This convention is used throughout the paper and SI.

Reviewer Comment:

The discussion after Eqn (2) is slightly misleading. The authors point out that for the reversible reaction the protein number changes by ± 1 , whereas for the approximated linear case it does not change. This is correct but it is not the main source of error in the approximation in my opinion. For large mean protein numbers, the main source of error in the approximation is that they replace a fluctuating rate of transition by an average rate; the fact that the protein number changes by ± 1 in the two reactions is less likely to be significant. I think a calculation of the FPT for transition between promoter states (see point below) can also address this point further.

Response:

We agree with the reviewer on this point. Thank you for pointing to us the connection with mean-field theory. We have changed the previous “local decorrelation assumption” by “mean-field assumption” and altered the discussion of this assumption accordingly after Eq. 2, in the Discussion Section and throughout the paper.

Reviewer Comment:

In the Applications of the LMA..(and in Section 4.6), the authors calculate the FPT for a switch between promoter states using their approximation scheme. However, this is done with the assumption that the initial state has zero proteins which is very restrictive and also not well justified. If we are in the steady-state limit, a more reasonable assumption is that the initial protein level is drawn from the steady-state protein distribution (or perhaps from the protein distribution conditional on promoter state) rather than setting it to zero. Furthermore, given the initial protein level, it should be possible (at least for the non-cooperative case) to calculate the FPT distribution exactly for Eqn. (27) (Section 4.6). The authors can simply write down the master equation for $P_{\{0\}}(n,t)$ (where n refers to the number of proteins at time t) and obtain the FPT distribution $P(t_{\{FP\}} = t)$ by differentiating wrt time (as they have on pg 12) and summing over all n . Now it should be possible to solve the master equation for $P_{\{0\}}(n,t)$ using the method of characteristics. It would be instructive if the authors can compare this exact solution with the solution given by the LMA approximation (pg 13). In particular, it would be interesting to note if there is a correlation between errors in the LMA approximation for the steady-state or time-dependent distributions with the corresponding errors in the FPT distribution.

Response:

This is an excellent idea and so we did as suggested. We calculated the FPT exactly in steady-state conditions for the feedback loop (see SI Section 4) and then compared it with the FPT distribution calculated using the LMA; this is now shown as Fig 7C. Indeed we find out that the error (as measured by the Hellinger distance) in the FPT distribution increases with σ_b , which is the parameter controlling the error in the LMA approximation for time-dependent distributions. This is now discussed in a paragraph after Eq. 7.

Reviewer Comment:

At different points in the text, the authors state that the LMA is exact when the condition of detailed balance is met. While this statement is correct, I feel it is somewhat misleading. For the cases considered, the feedback is irrelevant when detailed balance is satisfied, because the protein production rate is ρ regardless of the promoter state. So if, instead of using the promoter switching rate obtained using the LMA, the authors used any arbitrary switching rate in the linear model, they would still obtain the exact steady-state distribution (a

Poisson distribution with mean ρ). So it seems somewhat misleading to attribute this to the LMA and somewhat pointless to have a subsection in the supplementary material highlighting how the LMA result reduces to the exact result in this case.

Response:

We completely agree with this point and thus removed most references to detailed balance and the previous section treating it in the SI. An explanation along the lines suggested by the reviewer is now included in the second paragraph of Section 2.2: "It can also be easily proved that the LMA distribution is exact when the system is in detailed balance conditions. This is since in such conditions, $\rho_u = \rho_b$, which implies that the protein distribution is unaffected by the bimolecular reaction at the heart of promoter switching and hence the system acts as a linear GRN in this special case."

Reviewer Comment:

While the authors have looked at different models, the range seems to be restricted to models with only 2 promoter states. This is likely because analytical solutions for the time-dependent problem may not be tractable in the general case with more than 2 promoter states. However, the steady-state distributions are also of broad interest and, as noted by the authors, their approach is more straightforward and involves less assumptions if we focus on steady-state alone. So it would be useful if the authors could comment on the applications of their approximation scheme to models with multiple (> 2) promoter states.

Response:

We have now extended our LMA study to include feedback loops with explicit mRNA and protein dynamics, with nonlinear protein degradation propensity of the Michaelis-Menten type and as well to the multi-promoter toggle switch. In all cases the LMA gives accurate results. A discussion of these systems can be found at the end of Section 2.2, the associated graphs in Figs 5 and 6 and details of the LMA for these circuits can be found in the SI Sections 5, 6 and 7.

Reviewers' comments:

Reviewer #1 (Remarks to the Author):

Review: Linear mapping approximation of gene regulatory networks with stochastic dynamics

Zhixing Cao and Ramon Grima

The authors have revised the paper considerably and addressed most of my criticisms in the earlier review. I feel that the manuscript is now in much better shape, but before it can be recommended for publication, the authors should address the following points.

1. The authors mention on page 5 “*Note also that while all the three systems are composed of reactions with mass-action propensities, in certain limits they reduce to systems composed of effective reactions with non-mass action propensities e.g. under quasi-equilibrium conditions between promoter and protein, the model of a feedback loop with cooperative binding reduces to an effective model describing protein production with a Hill-type propensity [27, 28].*”

However even if non-mass action propensities can be approximated with a system of elementary reactions, these reactions will generally be bimolecular (like the enzyme-substrate reactions that lead to Michaelis-Menten kinetics in Figure 5). As these bimolecular reactions are not of “protein binding to a gene type”, applying LMA is not straightforward and heavy analysis is needed to find the effective rate constants for the linear dynamics. This analysis is shown for a simple network in Section 5 of *Supplementary Information* but it is unclear to me if such an analysis can be easily carried out for more general bimolecular networks. In such cases, applying LMA could be very difficult. I request the authors to comment on this issue.

2. In my previous review report I was concerned about the accuracy of the time-averaging assumption to obtain the time-dependent solution. In the revised version of the paper the authors provide a “*rigorous derivation of this assumption using the Baker-Campbell-Hausdorff series expansion*” which is reported in Methods section 4.5. While this derivation is good to have, I am still concerned because I don’t see why the higher-order terms in the time-discretization parameter Δ can be ignored on page 14. Note that for a fixed time T , the total number of discretization intervals N is inversely proportional to Δ (i.e. $N = T/\Delta$). In the second term (part B) of the Baker-Campbell-Hausdorff series expansion, each term in the sum is of order Δ^2 but the total number of terms is of order $N^2 = T^2\Delta^{-2}$, making the overall contribution of part B to be of order 1 which cannot be ignored. In fact, for large values of T , this second-term could even dominate the leading-term (part A).

In general for any two matrices A and B one can claim

$$\exp(A + B) = \exp(A) \exp(B)$$

if matrices A and B commute (i.e. $AB = BA$). So even if time is discretized over small-intervals $[T_i, T_{i+1}]$ of length Δ , and time-dependent $A(t)$ is assumed to be constant A_i over this interval, one **cannot** say that

$$\prod_{i=1}^N \exp(A_i \Delta) = \exp\left(\sum_{i=1}^N A_i \Delta\right)$$

unless A_i -s commute.

The authors should revise the derivation in Methods section 4.5 to address this problem.

1 Minor Issues

1. The title of the *Supplementary Information* document must be revised to match the new title of the paper.

Reviewer #2 (Remarks to the Author):

The authors have addressed my comments and the revised manuscript is significantly improved.

Reviewer #3 (Remarks to the Author):

The authors have adequately addressed all the points raised in the previous review. However, there is one point that needs to be addressed in the revised version. Eqn. 18 in the Supplementary Materials (section 4) is not exact as claimed. The point is that the probability distribution to be used in the calculation for "part B" is not the steady-state probability distribution of the feedback loop as claimed but in fact the probability distribution *just* after the switch from G^* to G . It is not clear that these two probability distributions are the same, indeed it seems unlikely that they are. In light of this, the analytical calculation is not exact as claimed, although it is likely to be an excellent approximation. It would be best to resolve this issue in the final version of the manuscript.

Response to reviewers for NCOMMS-18-00661B

We thank both reviewers for constructive comments that have really helped us to clarify some subtle points behind the LMA method. All changes in main text and SI are marked in red. Below are the detailed responses.

Reviewer 1

Referee Comment

However even if non-mass action propensities can be approximated with a system of elementary reactions, these reactions will generally be bimolecular (like the enzyme-substrate reactions that lead to Michaelis-Menten kinetics in Figure 5). As these bimolecular reactions are not of “protein binding to a gene type”, applying LMA is not straightforward and heavy analysis is needed to find the effective rate constants for the linear dynamics. This analysis is shown for a simple network in Section 5 of Supplementary Information but it is unclear to me if such an analysis can be easily carried out for more general bimolecular networks. In such cases, applying LMA could be very difficult. I request the authors to comment on this issue.

Author Response

Yes we agree with the referee and we have added the following paragraph in the Discussion section to discuss these points further:

“The LMA, of course, cannot possibly solve the master equations of all gene regulatory networks encountered in nature. In particular when the nonlinear GRN has also bimolecular reactions that are not of the protein-promoter type, the LMA mapping does not lead to a linear GRN though it is still a simpler GRN than the original one. In such a case, it is typically difficult to solve exactly the master equation of the simplified GRN. Likely, progress can then be made by replacing the bimolecular reactions (not involving protein and promoter) by an effective first-order reaction/s such that one has again an effective linear GRN. For example for GRNs for which the protein is catalyzed by an enzyme, the catalysis can be effectively modeled by a first-order decay reaction for the protein with a Michaelis-Menten rate (as shown in one of our examples). This additional linearization might not always be possible or else even if possible it might still lead to unsolvable or very difficult to solve master equations; this has to be ascertained on a case-by-case basis and no general statements can be made in this regard.”

Referee Comment

In my previous review report I was concerned about the accuracy of the time-averaging assumption to obtain the time-dependent solution. In the revised version of the paper the authors provide a “rigorous derivation of this assumption using the Baker-Campbell-Hausdorff series expansion” which is reported in Methods section 4.5. While this derivation is good to have, I am still concerned because I don't see

why the higher-order terms in the time-discretization parameter Δ can be ignored on page 14. Note that for a fixed time T , the total number of discretization intervals N is inversely proportional to Δ (i.e. $N = T/\Delta$). In the second term (part B) of the Baker-Campbell-Hausdorff series expansion, each term in the sum is of order Δ^2 but the total number of terms is of order $N^2 = T^2 \Delta^{-2}$, making the overall contribution of part B to be of order 1 which cannot be ignored. In fact, for large values of T , this second-term could even dominate the leading-term (part A).

In general for any two matrices A and B one can claim $\exp(A + B) = \exp(A) \exp(B)$ if matrices A and B commute (i.e. $AB = BA$). So even if time is discretized over small-intervals $[T_i; T_{i+1}]$ of length Δ , and time-dependent $A(t)$ is assumed to be constant A_i over this interval, one cannot say that

$$\prod_{i=1}^N \exp(A_i \Delta) = \exp\left(\sum_{i=1}^N A_i \Delta\right)$$

unless the A_i -s commute. The authors should revise the derivation in Methods section 4.5 to address this problem.

Author Response

These are very good points. We have completely revised the proof in Methods to address these concerns. Basically we find that actually it is much more convenient to work directly with the Magnus expansion (which can be seen as the continuous version of the Baker-Campbell-Hausdorff series expansion). The first term of this expansion is then exactly our LMA and the approximation error is the sum of the rest of the terms. We prove that the approximation is bounded in time and of order σ_b , the protein-promoter rate constant. Hence in the limit of small σ_b , the first term of the Magnus expansion (the LMA) dominates over all other terms, for all times. Finally we have a rigorous explanation of why the LMA works so well for all times and why the error is proportional to σ_b . The commutativity argument presented by the referee is correct however it is not cause for any concern. This is because the Lie bracket (measuring the degree of non-commutativity between the matrix A evaluated at different times) appears only in the second and higher-order terms of the Magnus expansion and these are, by the argument above, always much smaller than the LMA term, provided σ_b is small enough. The new derivation can be found in Methods Section 4.5 and also we have added a new section 8 in the SI to prove some results needed for the derivation.

Referee Comment

The title of the Supplementary Information document must be revised to match the new title of the paper.

Author Response

Thanks for spotting this – it is now changed.

Reviewer 3

Referee Comment

The authors have adequately addressed all the points raised in the previous review. However, there is one point that needs to be addressed in the revised version. Eqn. 18 in the Supplementary Materials (section 4) is not exact as claimed. The point is that the probability distribution to be used in the calculation for "part B" is not the steady-state probability distribution of the feedback loop as claimed but in fact the probability distribution *just* after the switch from G^* to G . It is not clear that these two probability distributions are the same, indeed it seems unlikely that they are. In light of this, the analytical calculation is not exact as claimed, although it is likely to be an excellent approximation. It would be best to resolve this issue in the final version of the manuscript.

Author Response

Thanks for pointing out this subtle fact. To check we computed Part B in Eq. 18 in the SI for three possible choices for the protein distribution: $P(n_p=n)$, $P(n_p=n|n_g=0)$ and $P(n_p=n|n_g=1)$ and compared with the SSA (see Fig. 3 in the SI). In all cases $P(n_p=n|n_g=0)$, the protein distribution conditional on the promoter being in state G^* gives perfect agreement with the SSA (within sampling error) while the other two choices do not. Intuitively it makes sense too because this is the distribution of proteins just before the switch from G^* to G , i.e., the distribution of proteins seen by the system when the reaction $G+P \rightarrow G^*$ becomes first possible. The results in Fig. 7C in the main text are practically unchanged however from before. We have revised the SI Section 4 to discuss this subtle point.

REVIEWERS' COMMENTS:

Reviewer #1 (Remarks to the Author):

The authors have successfully addressed the issues I had last time. I recommend publication of this paper in Nature Communications.

Reviewer #3 (Remarks to the Author):

The authors have addressed all concerns raised in previous reviews. I recommend publication of this manuscript.